# The Fossil Record and Diversity of Pycnodontiform Fishes in Non-Marine Environments

John J. Cawley [1] and Jürgen Kriwet [1,2,*]

1 Department of Palaeontology, University of Vienna, Josef-Holaubek-Platz 2, 1090 Vienna, Austria; cawleyj84@univie.ac.at
2 Vienna Doctoral School of Ecology and Evolution (VDSEE), University of Vienna, Djerassiplatz 1, 1030 Vienna, Austria
* Correspondence: juergen.kriwet@univie.ac.at

**Abstract:** Pycnodont fishes were a successful clade of neopterygian fishes that are predominantly found in shallow marine deposits. However, throughout their long 180 million year reign (Late Triassic–end Eocene), they made multiple incursions into both brackish and freshwater environments. This fossil record mostly consists of fragmentary dental material, but articulated specimens are known from Early Cretaceous lacustrine localities in Spain. This review article aims to document all non-marine occurrences of Pycnodontiformes throughout most of the Mesozoic and early Paleogene. This review highlights two interesting trends in the history of non-marine habitat colonization by pycnodonts: (1) a huge spike in non-marine occurrences during the Cretaceous; and (2) that most occurrences in non-marine localities occurred at the latest Cretaceous period, the Maastrichtian. The high number of colonization events within the Cretaceous lines up with extreme climatic events, such as high temperatures resulting in high sea levels which regularly flooded continental masses, allowing pycnodonts easier access to non-marine habitats. The increased presence of pycnodonts in brackish and freshwater habitats during the Maastrichtian might have played a role in their survival through the K/Pg extinction event. Freshwater habitats are not as vulnerable as marine ecosystems to environmental disturbance as the base of their food chain relies on detritus. Pycnodonts might have used such environments as a refuge and began to occupy marine waters after the K/Pg extinction event.

**Keywords:** pycnodonts; freshwater; brackish; Jurassic; Cretaceous; extinction; refuge





## 1. Introduction

Despite making up less than 0.3% of available global water masses, freshwater environments contain a staggering diversity of fish species, comprising approximately 25% of all known vertebrates [1] and 51% of all extant fishes. This high diversity in such a small area of habitat is due to high productivity, geographical isolation, and various physiographic processes [2,3] such as light [4], thermal zonation [5], and oxygen concentration [6]. However, the fossil record of freshwater fishes is of substantially poorer quality than that of their marine counterparts and thus, far less is known about the evolutionary history of many freshwater fish lineages. This relative paucity of fossil material is due to various environmental conditions such as strong currents and greater risk of drying out during arid seasons. Additionally, the freshwater fish fossil record is biased towards fish in lentic habitats (e.g., stillwater environments such as lakes and ponds) contrary to those from lotic habitats (moving waters such as rivers and torrents). This is powerfully demonstrated by the fossil record of two freshwater teleost lineages from the Neotropical region, i.e., Characiformes and Siluriformes. The fossil record of characiforms consists of fragmentary material such as isolated teeth and scales, testifying to their occurrence in sediments representing faster moving water, whilst fossil siluriforms are more commonly represented by complete fossils and are typically found in stillwater environments [7]. However, a factor we must

also consider is that siluriforms have denser and thicker bones than characiforms, making them more amenable to fossilization. Many neotropical siluriforms are also armored in some manner, whether they possess pectoral and dorsal spines or dermal plates, increasing their preservation chances further.

While many fish clades today are strictly freshwater (see [8]), there are many predominantly marine clades that have made incursions into, and adapted to, freshwater environments, such as members of Belonidae, Tetraodontidae, Acipenseridae, Mugilidae, and Gobiidae. Many other clades include diadromous taxa, i.e., those that move in between marine and freshwater environments during their life history [9]. Pertinent examples are members of Salmoniformes and Clupeidae, moving from marine environments into freshwaters to breed (anadromy), and freshwater eels of the family Anguillidae that migrate from marine environments as larvae and juveniles to rivers and lakes where they spend their adult life (catadromy).

In considering the history of any particular freshwater fish clade, it is rare that the clade in question originated in freshwater, and it has been noted by Cavin [10] that most basal taxa for any freshwater clade occurred in marine waters. Among 15 present-day families that had their origins in the Mesozoic, Patterson [11] only identified hiodontids, which are members of Osteoglossomorpha, as having originated and stayed in freshwater throughout their evolutionary history. During the course of their evolutionary history, many invasions of fishes into freshwater environments from lineages originating in marine waters have occurred. Some examples include Amiiformes [12], Ginglymodi [13], Acipenseriformes [14], and Coccolepididae [15].

Among the most successful lineages of fishes that lived during the Mesozoic were the pycnodontiforms or pycnodonts [16]. These fishes were considered to be stem teleosts by Nursall [17], but a more recent phylogenetic analysis by Poyato-Ariza [18] showed pycnodonts to be positioned, rather, on the stem of Neopterygii.

Pycnodonts are most commonly associated with shallow marine environments such as reefs and appear to be specialized for such complex, structured habitats due to their deep-bodied morphology and large medial fins (Figure 1) that are common to this order and are morphologically convergent with many extant coral reef fish families such as chaetodontids and balistids [19]. Cawley et al. [16] hypothesized that pycnodont fish diversity may be correlated with reef areas as their diversity was low when the area occupied by reefs was minimal but high when the reef area occupation increased. However, such a body plan can also be useful for other complex environments such as well-vegetated freshwater biotopes, and pycnodonts have seemingly repeatedly invaded brackish and freshwater environments multiple times throughout their history. Poyato-Ariza [20] consequently advised not to use pycnodonts as indicator taxa for any particular paleoenvironment, such as reefs, due to the occurrence of some in freshwater environments. Nevertheless, the true nature of these alleged freshwater pycnodonts has been since discussed. The review presented here aims to (1) compile all available pycnodont occurrences in brackish or freshwater deposits; (2) determine what this information tells us about their paleobiogeography and diversity through geological time; and (3) discuss the occurrence of fully freshwater-adapted pycnodonts.

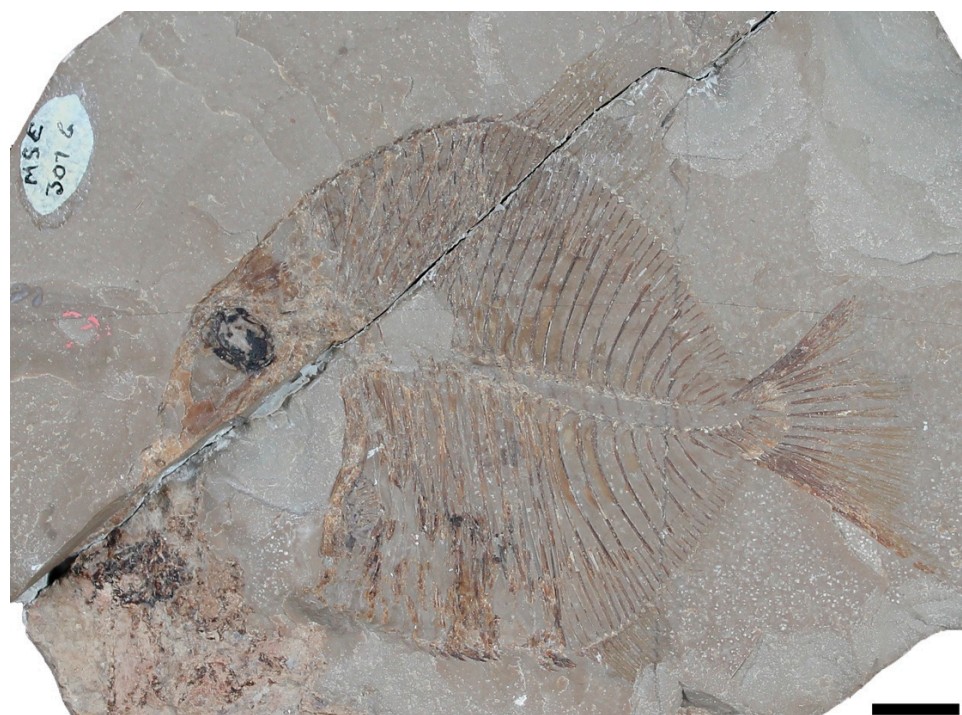

**Figure 1.** Holotype of *Ocloedus subdiscus* (MNHN MSE 301G) from the Early Cretaceous (Berriasian–Valanginian) El Montsec, Spain. This species represents one of the few articulated specimens of a pycnodont from a freshwater environment. Collection abbreviations: MNHN—Muséum national d'Histoire naturelle, Paris. Scale bar is 10 mm.

## 2. Non-Marine Occurrences

### 2.1. Middle Jurassic

Thailand

The earliest records of pycnodonts occurring in non-marine environments are difficult to accurately determine but can be traced back to the Middle Jurassic. The only definite record from this time interval, to our knowledge, is an isolated prearticular dentition from the Khlong Min Formation in Thailand that was presented by Cavin et al. [21] and identified as cf. *Gyrodus*. The Mab Ching locality has only been dated to be Middle-to-Late Jurassic based on charophytes [22,23] and fossil turtles [24]; unfortunately, more precise dating has not been inferred for this locality. Nevertheless, a Middle Jurassic age for the pycnodont remains seems likely since the Khlong Min Fm. is at the base of the stratigraphic sequence.

### 2.2. Late Jurassic

2.2.1. China

The Asian pycnodont record is very poor; so far, the only Late Jurassic Asian pycnodont that was reported from freshwater deposits is the little-known *Tibetodus gyroides* from Changtu in the Tibetan highlands [25,26]. This taxon is only known by an isolated vomerine dentition exposing five rows of regularly arranged teeth, and its age cannot be determined any further than being of the Late Jurassic age. Young and Liu [25] assumed this taxon to be a member of the family Gyrodontidae due to the dental similarities with *Gyrodus* such as the depressed pit in the occlusal surface of the teeth. However, they also argued that due to its widely separated geographic position from most other pycnodonts around this time, it is also possible for this taxon to belong to a different family. The morphology of the teeth actually differs from that of typical members of Gyrodontidae, supporting the latter interpretation.

### 2.2.2. North America

The only Late Jurassic records of freshwater pycnodonts from North America include a single broken tooth belonging to an undescribed pycnodontiform from the Morrison Formation (USA), which is of the Oxfordian–Tithonian age [27]. It was assigned to Pycnodontoidea due to its highly ornamented crown surface and oval shape. The preserved portion of the tooth displays at least seven small cusplets surrounding a median occlusal depression. Such ornate teeth are found in pycnodontiforms such as *Turbomesodon arcuatus* from the Berriasian (Early Cretaceous) of southern England and northwest Germany [28,29], as well as *Nonophalagodus trinitiensis* and *Texasensis coronatus* from the Albian period of North America [30,31] (see below for more information on these pycnodontiforms).

### 2.2.3. Africa

The only Late Jurassic African pycnodont non-marine record is the dubious *Congopycnodus cornutus* from the lacustrine Stanleyville Formation in the Democratic Republic of Congo. This taxon was based on the occipital region of the skull and a nuchal horn [32]. Previously, this formation was assumed to be Aalenian–Bathonian in age, which would have made this fish the oldest freshwater occurrence for pycnodonts. However, there is strong evidence that the Stanleyville Formation is actually Late Jurassic–Early Cretaceous (Kimmerdigian to Barremian–Valanginian) in age (see [33,34] for arguments). While the material of *C. cornutus* is fragmentary, the association of such bones together indicates that these remains represent the earliest known representative of the Coccodontoidea [35], a pycnodont clade containing highly derived armored pycnodonts such as Coccodontidae [36,37], Gladiopycnodontidae [35], and Gebrayelichthyidae [38], which are otherwise only found in the famous Cenomanian (Upper Cretaceous) marine limestone deposits of Lebanon. Whether this bizarre clade had its origins in freshwater environments or if *Congopycnodus* was an outlier within the early horned pycnodonts remains ambiguous at presents, requiring more and better-preserved material including the characteristic dentition. We consider the material not indisputable enough to unambiguously attribute it to a pycnodontiform at all.

An isolated vomerine dentition, referred to as cf. *Pycnodus*, was reported from the Mugher Mudstone of the Abay (Blue River) Gorge, northwestern Ethiopian plateau, which is of ?Kimmeridgian–Tithonian age [39]. The depositional environment of the Mugher Mudstone is interpreted as representing meandering river deposits on a coastal plain [40,41]. The tentative assignment to *Pycnodus* has to be considered incorrect since members of *Pycnodus* are exclusively restricted to the Paleogene [42]. This occurrence within a fluviatile environment on a coastal plain does not support the case for this pycnodont being a true freshwater fish, but it could represent a sporadic marine invader.

### *2.3. Early Cretaceous*

In contrast to the Jurassic, the records of freshwater pycnodonts increased during the Early Cretaceous, with a wide geographic distribution including some taxa which have fully preserved articulated specimens. This pattern is in good accordance with the fossil record of other fishes, for which the Early Cretaceous was seemingly an important time interval for adapting to freshwater environments (e.g., [10,43]).

### 2.3.1. Thailand

Small incomplete vomerine dentitions and isolated teeth assigned to cf. *Anomoeodus* were found from the Phu Phan Thong locality in the Sao Khua Formation in Thailand [21]. The teeth are of Hauterivian–Barremian age. Cavin et al. [21] assigned this material to cf. *Anomoeodus* due to the similarities of the shape and ornamentation of the teeth to those of *Anomoeodus*, which was widespread throughout the Cretaceous [44]. Of particular interest is that all specimens of cf. *Anomoeodus* found in this locality are tiny, suggesting that this pycnodont travelled into freshwater environments to breed and that their young would have developed in freshwater habitats before returning to sea in adulthood [45].

### 2.3.2. United Kingdom

Pycnodont records from non-marine deposits of the United Kingdom come exclusively from England. Hornung [29] presented a revision of Early Cretaceous pycnodonts not only from Germany (see below) but also from England, which includes various fragmentary prearticulars and vomers previously referred to as *Coelodus*. According to this author, all specimens can either be assigned to *Turbomesodon arcuatus* (Figure 2) or *T. laevidens*, both occurring in the Middle Purbeck limestone, which is Berriasian and Berriasian–Valanginian in age, respectively; to *T. microdon* from the Grinstead Clay Formation of the Valanginian; or to *T. multidens* from the Sevenoaks locality of the Weald Clay Group of the Hauterivian–Barremian. Hornung [29], moreover, infers that "*Pycnodus*" *mantelli* and *T. arcuatus* may be synonymous. If this is true, then the location of *T. arcuatus* in southern England and "*Pycnodus*" *mantelli* in Germany could be an indication that this taxon might be amphidromous due to the lack of a hydrographic connection between the continental Wessex and the Lower Saxony basins. However, both a better quality and a higher quantity of specimens from both localities will be necessary in order to confirm such a hypothesis.

Sweetman et al. [46] reported premaxillary, dentary, vomer, and prearticular bones with teeth from the Wessex Formation (Barremian) from southern England and assigned them to *Coelodus* sp. However, Cooper and Martill [47] observed that the morphology of the teeth better resembles that seen in *Ocloedus* [48]. An additional, still-unidentified pycnodont in the same locality, referred to only as Pycnodontiformes indet., is represented by fragmentary vomers and prearticulars along with isolated teeth.

The occurrence of the pycnodont *Coelodus* was reported from two layers in the Berriasian Middle Purbeck beds of Durlston Bay in southern England [26]. These remains were obtained from the Cherty freshwater layer of the Lulworth Formation and the inter-marine layer of the Durlston Formation, the paleoenvironment of the former being a series of freshwater ponds and marshes and the latter being a freshwater–brackish lake based on salinity measures inferred from mollusk shells and evaporite data [49].

However, the taxonomic assignment to *Coelodus*—or to a pycnodont at all—remains dubious because no description nor any figure of this material has been provided up to now, thus not allowing for an unambiguous identification of this pycnodont fish; it was only listed in a table. Thus, unless the specimens related to this taxonomic assignment can be found or further material from this locality can be recovered, these specimens should be considered dubious at the best.

Another Early Cretaceous freshwater pycnodont record based on isolated teeth comes from the Pevensey Pit of the Ashdown Brickworks quarry of Bexhill in southern England, which is Valanginian in age [50,51]. Even though these authors did not provide any taxonomic assignment, we tentatively assign them here to *Ocloedus* because of the characteristic tooth morphologies. These teeth, alongside material belonging to a diverse fauna of sharks, teleosts, amphibians, and dinosaurs, would have been deposited in a rapidly flowing river due to such material being reworked together with sediments along pebble shorefaces [52,53]. Additional evidence for this environment being freshwater can also be seen in the predominance of certain genera of ostracods and bivalves such as Cypridea and *Neomiodon*, respectively [54].

### 2.3.3. Germany

The non-marine Early Cretaceous fossil record of Germany is, according to our current knowledge, restricted mainly to the Berriasian of northern and northwestern Germany. Due to the lack of complete or taxonomically informative fossil material, most of these specimens are referred to as Pycnodontiformes indet. Isolated teeth were recovered from various lacustrine facies of the Berriasian Bückberg Group, such as Sehnde and Wätzum [Fuhse Fm.] [55,56] and Egestorf in Lower Saxony [Isterberg Fm.] (late Berriasian), which were initially assigned to *Pycnodus mantelli* by Struckmann [57] but lack any description or illustration; Barenberg near Borgholzausen in Northrhine-Westphalia [either Oesede or Isterberg formations], previously referred to as *Gyrodus mantelli*, which only were

mentioned but not illustrated by Dunker [58]; and finally Lobber Ort on the Island of Rügen [59]. These remains have to be considered ambiguous for as long as no further information is available.

Prearticulars assigned to "*Pycnodus*" *mantellii* (aka *T. arcuatus*) were also reported from various Berriasian sites in Lower Saxony [58] and North Rhine-Westphalia, northern Germany [60]. Vomers and isolated teeth belonging to *Turbomesodon* cf. *arcuatus* were also discovered in North Rhine-Westphalia [29,60–62]. Another specimen of this taxon represented by a vomerine dentition was reported from an unknown locality in the Isterberg Formation of the Bückenberg Group [29]. Kriwet [63] also figured a specimen of this taxon as "*Coelodus*" sp. from northern Germany [[63]; fig 28c].

While the paleoenvironments of all these Berriasian localities are interpreted as being lacustrine, occasional marine influences cannot be completely rejected.

### 2.3.4. Belgium

While the locality of Bernissart in Belgium is most famous for the *Iguanodon* remains found at the bottom of a pit, a staggering number of actinopterygian specimens were unearthed from this locality, amounting to nearly 3000 specimens containing 17 species belonging to 12 genera [64,65]. The fossil remains were found in lacustrine clays, which are defined as the Sainte-Barbe Clays Formation [66,67]. The environment of Bernissart would have been lacustrine to swampy [68–71], and using palynological dating, it is estimated to be of late Barremian to early Aptian age [72–74]. The only pycnodont taxa found so far in this locality are some complete but imperfectly preserved specimens of *Turbomesodon bernissartensis* [75,76].

### 2.3.5. France

An assortment of isolated teeth of various morphotypes from the Berriasan locality of Champblanc Quarry in Cherves-de-Cognac, France, can be referred to as Pycnodontiformes [77]. These teeth are commonly found in the upper levels of the Champblanc section, which are characterized by lacustrine limestones [78,79]. The most common morphotype found is similar to the morphology seen in *Gyrodus*, with a papilla-like structure present in the middle of an occlusal pit, and Pouech et al. [77] subsequently considered this morphotype as aff. *Gyrodus*.

The only other morphotype that could be identified with a particular genus are teeth with two rounded tubercles on each end of the tooth crown with a groove in between. Such tooth morphology is similar to that seen in *Arcodonichthys* [80]. Every other morphotype from this locality cannot be assigned to any genus or species, with the most granular assignment given being that of the family Pycnodontidae.

The large amount of different and diverse morphotypes in the Berriasan locality of Champblanc Quarry nevertheless indicate that a diverse pycnodont fauna occupied this habitat that probably underwent niche partitioning in order to exploit different resources efficiently [16,47,81]. Another interesting observation by Pouech et al. [77] was the distribution of pycnodonts within the sedimentary series of Cherves-de-Cognac compared to ginglymodians, another actinopterygian clade with many durophagous representatives. Pycnodonts are more commonly represented in the upper levels of the sedimentary series, which are dominated by lacustrine limestones, leading the authors to propose a lacustrine life environment for pycnodonts. Lentic environments such as lakes might have been more ideal for the typical laterally flattened and dorsoventrally high body of pycnodonts as their morphology already permits them to maneuver in shallow, complex environments without expending energy fighting the current. In contrast, isolated teeth of ginglymodians such as *Scheenstia mantelli* dominate the actinopterygian assemblage in the lower levels. The environment of this lower series is confirmed by geochemical analyses to be upstream waters close to the lagoon where these teeth were deposited [82]. This pattern of pycnodonts and ginglymodians occupying different habitats mirrors that seen in marine environments,

as shown by the separation of these two groups from the same locality in a body shape morphospace occupation [16].

2.3.6. Spain

The fossil record of non-marine pycnodont occurrences of Spain consists of not only abundant isolated dental remains but also many articulated skeletal remains (Figure 1), including holomorphic specimens.

A large abundance of isolated vomerine and prearticular tooth plates belonging to the pycnodont *Arcodonichthys pasiegae* were recovered from the Vega de Pas Formation (Hauterivian–Barremian) in the Cantabrian basin, northern Spain [80]. Sedimentation of this formation occurred in a lacustrine setting. *Arcodonichthys pasiegae* is easily identifiable due to the strongly curved anterior and posterior border of the vomerine teeth with swollen lateral ends. Due to the large number of specimens of this taxon found (over 100), some interesting aspects of its ecology can be inferred. The measurements of the tooth plates (see table in [80]) suggest a small pycnodont, close in size to taxa such as *Stemmatodus* [83], with an estimated size range of 4–7 cm. Tooth wear is extensive even in smaller specimens, and the inner layer of dentine can be exposed in larger specimens [[80]; fig 1c]. This dental evidence, along with the abundance of bivalves and gastropods found within the same fossil layers [84] as *Arcodonichthys* suggest that this species was particularly specialized to feed on hard-shelled prey in comparison to many other pycnodonts, which were hypothesized to be more generalized durophages with a broad range of potential prey items, as advocated by Poyato-Ariza [20] and Darras [85]. However, Kriwet [86] already hypothesized that pycnodonts were highly specialized feeders.

Several Lower Cretaceous localities in Eastern Spain yielded additional pycnodont fish material, including complete skeletons. All these localities are situated in the Mesozoic Iberian Basin, an intracratonic basin situated at the eastern margin of the Iberian plate. The development of the Lower Cretaceous sub-basins of the Iberian Basin is related to the anticlockwise rotation of the Iberian Plate during the Mesozoic and extensional tectonic events [87].

Two of these famous localities are La Pedrera de Santa Maria de Meià (= La Pedrera de Rúbies) near Vilanova de Meià and La Cubrua in the Sierra de Montsec, Province of Lérida, northeastern Spain (e.g., [88,89]). Originally assumed to be of Kimmeridgian age [90], Peybèrnes and Oertli [91] and Brenner [92] attributed a late Berriasian to early Valanginian age to the fossiliferous strata. The fish fauna include chondrichthyans (only hybodonts [93–95]), an actinistian, and many actinopterygians [96–100]. Pycnodonts are abundant in the Sierra del Montsec and are represented by adult and juvenile specimens of a single species, *Ocloedus subdiscus* [48,89,101]. The depositional area of Montsec is subject to dispute. While some authors [102,103] interpret the depositional area as a freshwater environment based on terrestrial faunal elements such as plants, insects, and tetrapods, the occurrence of agglutinating foraminifers, acritarchs, and brackish ostracodes indicate a marine environment [104]. This interpretation is also supported by the composition of the fish fauna [105]. The paleogeographic situation of the fossiliferous strata also supports a coastal, intertidal environment, which is advocated here.

The syncline of Galve in the Province of Teruel in northeastern Spain belongs structurally to the Lower Cretaceous sub-basin of Aliaga. The basin is built up of about 1000 m of Jurassic and Lower Cretaceous sediments and has yielded ca. 60 vertebrate localities ranging from the Jurassic to Early Cretaceous in age (for reviews of the vertebrate fauna, see [106,107]). The Early Cretaceous Aliaga sub-basin can be regarded as one of the most productive vertebrate localities of the Early Cretaceous age on the Iberian Peninsula. Pycnodonts are represented by abundant isolated dentitions, including both large and small specimens of the same taxon, indicating the presence of both juveniles and adults. Indeterminate pycnodontiforms were reported by Sánchez-Hernández et al. [107] from the upper part of the Berriasian–Barremian El Castellar Formation, which is interpreted as being an open lacustrine environment with distributary channels and a lagoonal system near

the coast based on the presence of bioclastic limestones, sand beds, charophyte remains, and root marks [108]. Soría and Meléndez [109] reported the presence of cf. *Proscinetes* sp. and *Coelodus* sp., as well as unidentifiable forms in the lower Barremian Camarillas Formation, which was interpreted by these authors as a fluvial environment. The presence of *Anomoeodus* or a similar taxon is indicated by the presence of an isolated premaxilla, which is characterized by three incisiform teeth (JK, pers. obser.). Lagoonal conditions in the syncline of Galve are supposed to have been established during the upper Barremian Artoles Formation, in which pycnodont dental remains are common (JK, pers. obser.). In contrast to this environmental interpretation, the selachians and associated actinopterygian fauna seems to favor an environment with at least sporadic connections to the open marine realm in the east [110].

Additional isolated dental remains of three different pycnodonts, including two different pycnodontids of uncertain affinities (probably representing *Ocloedus*) and *Anomoeodus nursalli*, were described by Kriwet [44] from coaly limestones (carbonates and clastics interbedded with coal seams of 60 to 65 cm thickness) of Uña in the Province of Cuenca, eastern Spain. The coals were described as brown coals or lignites [111–113]. They belong to the Huérgiuna Formation, which is early Barremian in age [114]. The depositional area is generally interpreted as freshwater. In addition to the pycnodonts, still-undescribed actinopterygians and a diverse tetrapod fauna were derived from one of these coals. Kriwet [44] hypothesized that the coals of Uña were accumulated at the margin of the Bay of Cuenca, which was connected to the marine realm in the east. Accordingly, continental faunal and floral contents were washed into this fast-changing area of paralic coal formation or lived at the edge of the bay. The pycnodonts were thus probably not primary inhabitants of this restricted and probably fast-changing environment.

Although the vast majority of the Spanish Early Cretaceous non-marine pycnodont fossil record seemingly consists of fragmentary material such as isolated teeth and jaws, some complete pycnodont specimens from freshwater deposits were nevertheless recovered [89,115–117]. The most notable, in addition to *Oeclodus subdiscus* from Montsech (see above), are *Stenamara mia* and *Turbomesodon praeclarus* from the lacustrine deposits of Las Hoyas in the Province of Cuenca, eastern Spain, which also is of Barremian age [76,118]. This locality was shown to represent a freshwater environment without any marine influence based on many factors, including paleoecology, paleogeography, sedimentology, stratigraphy, taphonomy, and extensive isotope analysis of the fish fossils [119]. Both co-occurring pycnodonts are members of the Pycnodontidae, the most specious lineage of pycnodonts, and are considered the first unambiguous record of non-marine pycnodonts to be discovered [119]. *Stenamara* is both the rarest and smallest of the two Los Hoyas pycnodonts, with only two specimens known [117] and its standard length being 73 mm. While *Stenamara* is comparatively poorly preserved, it can still be ascertained to include an adult specimen due to the presence of bifurcation in the lepidotrichia [100], and its extremely deep body (maximum body height/standard length being 124.5%) indicates it to have been a slow swimmer but well adapted to maneuver in complex environments such as the vegetated shallows of the lake of Las Hoyas. *Turbomesodon praeclarus*, originally identified as *Macromesodon* aff. *M. bernissartensis* [120], conversely, was obviously much more common in Las Hoyas, with 25 or more complete or partially preserved specimens along with isolated dentition [76]. Like *Stenamara*, it was a deep-bodied fish well adapted to maneuver in structured habitats. The discovery of small specimens of *T. praeclarus* from the Las Hoyas sediments indicate the presence of juveniles, which is considered additional evidence that these fishes were fully freshwater adapted as they evidently also reproduced in the lake [76,120].

Poyato-Ariza [117] speculated that the Las Hoyas pycnodonts might have been more herbivorous than what is typically assumed for the group as the usual pycnodont prey, such as mollusks, is not present in the Las Hoyas locality, and crustaceans such as crayfish are too large for the small-gaped pycnodonts to feed on and break apart. The charophyte alga *Clavatoraxis robustus* was in great abundance in Los Hoyas [117] and was reinforced

with spine-cell rosettes, which Martín-Closas and Diéguez [121] interpreted as a defence mechanism against herbivory. Both *Stenamara* and *Turbomesodon*, with their powerful molariform teeth and nipping incisiform teeth, might have been able to tackle such well-defended prey. This herbivory hypothesis is more plausible than initially assumed due to adaptations to plant eating being found in other pycnodonts including bicuspid teeth in the Late Triassic *Gibbodon cenensis* [122], *Nursallia tethysensis* [123], and *Thiollierepycnodus wagneri* [124]. A large premaxillary tooth with multiple cusps similar to that seen in modern herbivorous teleosts such as marine acanthurids and freshwater cichlids can also be seen in the Paleocene *Pycnodus multicuspidatus* from Moroccan deposits [125]. A tooth microwear analysis by Baines [126] on the jaws of the Jurassic pycnodont *Gyrodus planidens* revealed that the orientation of dental wear patterns on the teeth show that it was capable of propalinal or horizontal jaw movements, which is similar to the behavior seen in mammals. These jaw movements would have been effective for processing plant material in contrast to the vertical crushing actions more typically used to access hard–shelled prey items. The comparatively high number of *T. plaeclarus* specimens in particular could be used to test such dietary hypotheses in the future.

Some interesting trace fossils at Las Hoyas can possibly be attributed to pycnodonts. These horizontal trails consisting of a sinusoidal wave are referred to as the ichnospecies *Undichna unisulca* [127] and are preserved as epichnial grooves or hypichnial ridges. These trace fossils are more commonly preserved in freshwater environments and are a decent indicator of paleoenvironmental setting [127]. The producer of these trace fossils would need to possess a large caudal fin with no overlap with the anal fin to produce such traces in the sediment. There are currently no fishes known from Las Hoyas that have such a caudal fin morphology without another fin leaving occasional traces. Pycnodonts typically have large anal fins, which would fit in with the form of the trace fossils left behind, and so were posited by Gibert et al. [127] to have been the most likely producers of *U. unisulca*. *Turbomesodon praeclarus*, being the most common of the two pycnodont taxa found at Las Hoyas, is the most likely producer of these traces. Nevertheless, the paleogeographic situation of Las Hoyas close to the marine realm during the Barremian does not preclude occasional marine ingressions.

In addition to these well-known vertebrate sites, many other Early Cretaceous localities on the Iberian Peninsula, which are supposed to represent freshwater settings, yielded isolated dental remains of pycnodonts, such as from the Berriasian of Lérida, eastern Spain (*Proscinetes* sp. cf. *P. bernardi*); the Barremian of Aguilón near Zaragoza, eastern Spain (*Coelodus* sp.); the Aptian of Cintorres near Valencia and Mirambel in Teruel, eastern Spain (*Anomoeodus complanatus*); the Aptian of Morella, Valencia, eastern Spain (*Coelodus* sp. aff. *C. soleri*); the Albian of Ceceda, Asturias (undetermined pycnodont); the Albian of Condemios de Abajo, Castilla-La-Mancha, central Spain (*Anomoeodus* sp. aff. *A. muensteri*); and the Albian–Cenomanian of the area of Girona in Catalonia (*Coelodus soleri*) [93,128–131]. No pycnodont fish remains from non-marine deposits of the Early Cretaceous age have been reported from western or southern Spain or even Portugal to date. Nevertheless, this review emphasizes the incomplete knowledge of these fishes from the Early Cretaceous period of the Iberian Peninsula, both in terms of taxonomy and environmental distribution.

### 2.3.7. North America

The currently known non-marine Early Cretaceous record of pycnodonts from North America is seemingly restricted to the Aptian–Albian. Oreska et al. [132] reported five partial vomers, which they assigned to aff. Pycnodontidae indet. This material was collected from two vertebrate microfossil beds at the Little Sheep Mudstone Member of the Cloverly Formation of Wyoming, which has been dated to the Aptian period [133]. These microfossils were accumulated in concentrations within clay and mudstones, and the lack of wear on these fossils indicate low transport and a low energy depositional environment [134], which, along with fine-grained sediments and weathering volcanic ash, can be interpreted as a lacustrine environment [135].

Pycnodont remains were also reported from the Holly Creek Fm., which is part of the Trinity Group in Arkansas, and which is dated to the Albian [136]. The depositional environment is suggested to be a coastal plain, being a mixed-water-to-brackish environment based on stable isotope work performed on turtle shells [137]. This formation contains two pycnodonts: Pycnodontiformes indet. and *Anomoeodus caddoi* [136]. *Anomoeodus caddoi* is represented by one isolated tooth and two prearticular toothplates. A singular branchial and oral tooth are the only materials representing Pycnodontiformes indet.

The greatest number of North American non-marine pycnodont occurrences during the Early Cretaceous can be observed in northern Texas [30,138–141] and consist exclusively of isolated dental remains. Based on these remains, several taxa were identified, such as *Texasensis coronatus*, from the Paluxy Fm. of the Huggins and Pecan Valley Estates sites of northern Texas (Albian), which consists of a series of sandstones and shales representing a fluvial meander belt to a coastal barrier and deltaic facies [142]. This species was originally referred to as *Callodus coronatus* by, e.g., Thurmond [30] and Winkler et al. [139,140], but was transferred to *Texasensis* by Özdikmen [31] because the name of *Callodus* had already been taken by a beetle.

Another pycnodont that also occurs in the Pecan Valley site is "*Macromesodon*" *dumblei* (most likely a member of *Turbomesodon*), while *Nonaphalgodus trinitiensis* was described from the Albian deposits of Butler Farm [143], which is also considered to represent a freshwater pycnodont. The pycnodont *Thurmondella estesi*, which was originally ascribed to the genus *Paramicrodon*, which, notably, represents a synonym of a genus of flies [144], is known from two Texas localities spanning two different geological stages, i.e., Albian and Aptian, both belonging to the Paluxy Fm. [140]. The fossils of this species were recovered from deposits which are in close association with thin, variably developed bituminous coal and plant-fragment-bearing mudstones, indicating continental influence [140]. Other taxa recovered from the Albian non-marine localities of northern Texas include "*Paleobalistum*" *geiseri*, "*Proscinetes*" *texanus*, and "*Coelodus*" sp. [140].

### 2.3.8. South America

South American Early Cretaceous non-marine pycnodont records come from a fossil fish assemblage of the Açu Formation, which is located in the Potiguar Basin in Ceará, northeastern Brazil. The deposits of this formation were interpreted as a meandering riverine environment with some tidal influence [145]. Precise dating of this formation has been unsuccessful so far due to the lack of stratigraphically useful and diagnostic fossils, but an Albian–Cenomanian age was suggested based on palynological analyses [146,147]. The more recent stratigraphic data provided by Arai [148,149] specified the age of the Açu Formation as Albian in age.

Isolated teeth and a partially preserved vomerine and prearticular dentition of pycnodonts were recovered from the Açu Formation [150] and assigned to "Pycnodontiformes incertae sedis" due to the lack of more complete and taxonomically diagnosable material. These specimens are likely to represent a new genus as they differ from all other known Brazilian pycnodont taxa in the shape and ornamentation of the teeth, which are present as a series of crenulations on the margin of the tooth (Figure 2) [150].

### 2.3.9. Africa

Fragmentary prearticular and isolated teeth from the Albian Jebel Boulouha North locality in Tunisia were assigned to tentatively aff. *Gyrodus* sp. by Cuny et al. [151]. Sedimentation of the fossiliferous layers were interpreted as a terrestrial carbonate-rich environment [152]. The teeth display an apical depression, which is surrounded by crenulated margins similar to what is seen in European specimens also assigned to aff. *Gyrodus* sp. from the Hauterivian by Kriwet and Schmitz [153]. This similarity prompted Cuny et al. [151] to assign this material to a similar taxon, pending better-preserved material.

Another Tunisian specimen from the Albian described by Cuny et al. [151] is an isolated tooth identified as Pycnodontiformes indet., which was recovered from the Oued

el Khil assemblage. The depositional area of this assemblage was interpreted as a brackish environment due to the mix of freshwater and marine influences found in deposition [152,154].

The Kem Kem Group of Morocco is one of the most famous Cretaceous freshwater localities, being Albian–Cenomanian in age. While better known for hosting a multitude of giant vertebrate taxa, such as the theropod dinosaurs *Spinosaurus* [155–157] and *Carcharodontosaurus* [158], the sawfish *Onchopristus* [159,160], and the coelacanth *Mawsonia* [161], the Kem Kem Group also provided a diverse pycnodont fauna [47]. All of the known pycnodont taxa from this locality are represented by fragmentary dentitions and isolated teeth, which are poorly preserved probably as result of the high-energy riverine environment that characterizes the sedimentary setting of the Kem Kem. The corresponding pycnodont fauna is of note for two things in particular: (1) it represents the oldest pycnodont record from Morocco; and (2) it represents the most diverse record of freshwater pycnodont assemblages known in the fossil record, with at least four genera being represented. This assemblage also allowed for the identification of a second species of *Neoproscinetes*, *N. africanus*. This species shares with *N. penalvai* from the Early Cretaceous of Brazil similar circular-to-subcircular medial vomerine teeth and similar morphologies, numbers, and arrangements of teeth on the prearticular (Figure 2). *Agassizilia erfoudina*, an endemic taxon for the Kem Kem Group, is similar to *Iemanja* from the Early Cretaceous period of Brazil in that the prearticular teeth are small and irregularly arranged, but they are only restricted to the lateral margin in *Agassizilia* (Figure 2). This suggests that it may have also fed on soft-shelled crustaceans, as was suggested for *Iemanja* [47]. The other two pycnodont taxa are not well enough preserved to allow for a definitive taxonomic assignment but were identified as cf. *Macromesodon* and cf. *Coelodus* due to their sharing many similarities with the aforementioned genera of pycnodont fishes. Cooper and Martill [47] hypothesized that the high diversity of pycnodonts in the Kem Kem Group could be due to niche partitioning as each taxon found has a unique arrangement of teeth, which, combined with the few stomach contents of pycnodonts that have been preserved being monospecific [86], suggests high specialization of each genus to different prey items. Additional evidence for such niche partitioning was found through the results of a morphospace analysis concerning pycnodont lower jaws [16].

*2.4. Late Cretaceous*

2.4.1. China

Zhou et al. [162] documented a hitherto unknown pycnodont, *Xinjiangodus gyrodoides*, which is represented by a fragmentary vomerine and prearticular dentition coming from the sandy mudstones of the Donggou Formation at the southern margin of the Junggar Basin in Xinjiang, China. The exact age of this formation is problematic, with some authors indicating a Coniacian–Santonian age [163] and others restricting it to the Maastrichtian [164]. More recent work based on fossil ostracods attributes a Coniacian–Campanian age to the Donggou Formation [165,166]. The sedimentary environment of the fossiliferous site of this pycnodont is indicative of shallow lake deposits. A characteristic feature of *X. gyroides* distinguishing it from other pycnodonts is the large spacing between its teeth, as well as the presence of an apical depression on the tooth crowns with radiating wrinkles ([162], Table 1). An isolated opercular bone was also assigned to this taxon. However, the shape and size of the fragmentary opercular bone is far too wide and large for pycnodonts, and it is more probable that it belongs to a teleost. Alternatively, it could be the remains of a preopercular bone of a pycnodont as that bone is typically more expanded in pycnodonts, but this specimen is too fragmented to genuinely identify this bony element to any pycnodontiform, let alone *X. gyrodoides*.

2.4.2. India

Complete pycnodont skeletons representing a single species, "*Pycnodus*" *lametae*, have long been known from the Late Cretaceous Lameta Formation of central India (Province of Maharashtra) [167,168]. The Lameta Fm. is part of the infratrappean beds, which

were deposited below the Deccan Traps. These traps represent the deposits of active volcanic flows during the latest Cretaceous (Maastrichtian) period found in association with sedimentary beds of continental origin. The infratrappean beds of the Lameta Fm. were interpreted as palustrine/lacustrine depositional environments in an alluvial plain setting under semiarid climatic conditions based on paleontological and sedimentological evidence [169–172].

An additional incomplete specimen of "*Pycnodus*" *lametae*, consisting of a crushed skull and missing caudal region, was described from laminated clays of the same area from which the specimen of Woodward [167] originated at Dongargaon by Mohabey and Udhoji [173]. These authors mistakenly assigned this specimen as the holotype despite the holotype being the specimen described by Woodward [167]. The assignment of this pycnodont to *Pycnodus* is problematic due to the lack of a post-parietal process (reported in [42]), so revision of this material is necessary for taxonomic purposes.

Isolated teeth of various tetraodontiforms, which were initially assigned to *Stephanodus libycus*, *Indotrigonodon ovatus*, *Pisdurodon spatulatus*, and *Eotrigonodon wardhaensis* were also reported from the Lameta Fm. [174,175]. Prasad [168] provided a detailed review of the vertebrate fauna of the Deccan Traps and reassigned these teeth to an indeterminate pycnodontid. The teeth of *S. libycus* were interpreted as branchial teeth, whereas the teeth of *I. ovatus*, *P. spatulatus*, and *E. wardhaensis* were interpreted as oral teeth [168]. Shome and Chandel [176] reported the presence of branchial teeth, which were assigned to *Pycnodus* sp. from the Papro Fm. of Lalitpur District, Uttar Pradesh, which is dated to the Maastrichtian based on the vertebrate microfauna fossil record. The oral teeth of "*Pycnodus*" sp. and branchial teeth, named *Stephanodus libycus*, were also discovered from predominantly freshwater, lacustrine, and palustrine (swamp) deposits of the Maastrichtian Lameta Fm. in the Province of Madhya Pradesh [177,178].

Many additional records of pycnodonts are known from intertrappean beds. In contrast to infratrappean beds, intertrappean beds were deposited during the dormancy stages of volcanic activity and are thus intercalated within the lava flows. These depositions are generally considered to have been freshwater lacustrine paleoenvironments that would have occupied low-lying areas on the surface of lava flows when volcanic flow had ceased.

Khosla et al. [179] reports the presence of a branchial tooth assigned to an indeterminate pycnodontid from an intertrappean section of Kisalpuri, which also contains freshwater ostracods. Teeth named *Pycnodus bicresta* were reported from the Naskal locality of the Andhra Pradesh Fm., which was interpreted as an alkaline lake in a floodplain setting based on sedimentlogical, paleontological, and taphonomic evidence [180,181]. Since these authors considered pycnodonts to be an exclusively marine group, they assumed that *P. bicresta* had migrated into this lake through river channels and became subsequently entrapped.

Also, from Naskal, a specimen of "*Pycnodus*" *lametae* was reported to have been recovered, which, unfortunately, was lost during preparation [182]. It seems that "*Pycnodus*" *lametae* is a common freshwater taxon in India, as Rana [183] has reported specimens from several localities including Asifabad, Nagpur, and Rangapur. As well as "*Pycnodus*" *lametae* [184], two other pycnodont taxa, *Pycnodus bicresta* [168,185], which was originally assigned as an indeterminate pycnodontid [175,184], and *Pycnodus* cf. *P. praecursor* [183], are known from the Asifabad Fm., the latter restricted to infratrappean beds while the other pycnodont specimens are also found in intertrappean beds.

Another unidentified pycnodontid, represented by a prearticular bone, branchial, incisiform, and molariform oral teeth, is known from the Maastrichtian Piplanarayanwar of the Chhindwara District, Madhya Pradesh [185]. This locality was interpreted to have a nearshore, coastal plain environment due to the abundance of the myliobatid batomorph *Igdabatis indicus*. Diverse mixes of non-marine and marine taxa at the Asifibad and Nagpur sites also indicate some marine influences [184,186,187].

Two other Indian localities of the Late Cretaceous to Paleogene age provided additional isolated pycnodont teeth, which all were identified as *Pycnodus* sp. These include the

estuarine deposits of the Maastrichtian–Selandian Fatehgarh Fm. of Rajasthan [188,189], and the Maastrichtian–Danian intertrappean beds at the Jhilmili locality, Chhindwara District [190], in which the presence of *Pycnodus*—together with *Lepisosteus*—remains indicate a fluviolacustrine environment. It nevertheless should be noted that all Late Cretaceous pycnodont remains assigned to *Pycnodus* have to be considered as different taxon since *Pycnodus* is not known before the Cenozoic (compare [42]).

### 2.4.3. Austria

Isolated teeth and a fragmentary right prearticular were reported from the Turonian Schönleiten Formation in SE Austria (Steiermark) and assigned to *Coelodus plethodon* by Schultz and Paunović [191]. This species was originally described by Arambourg and Joleaud [192] from the marine Campanian of Niger (Africa). Ősi et al. [193] argued that the Austrian material is too fragmentary to assign it unambiguously to any particular genus, with which we agree here. Consequently, these remains have to be considered as Pycnodontidae or even Pycnodontiformes indet. The specimen was recovered from grey shales which represents a brackish, estuarine environment [194–196].

### 2.4.4. Hungary

The pycnodontid cf. *Coelodus* was reported from the Santonian Csehbánya Fm. of Iharkút, Hungary, based on isolated dental remains [197–199] (Figure 2). The prearticulars were initially considered likely to represent two different morphotypes by Ősi et al. [200], but Szabó et al. [199] nevertheless assigned all the Iharkút material to a single taxon until more complete material becomes available for a definite identification.

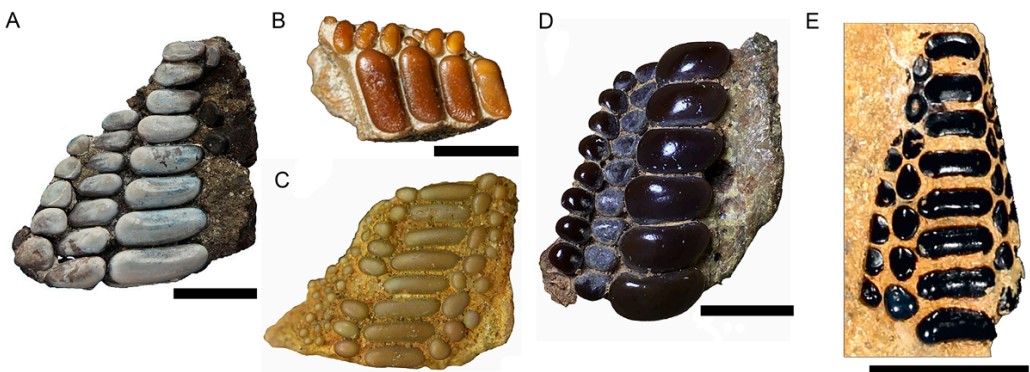

**Figure 2.** Examples of jaw fragments of pycnodont taxa from non-marine localities. (**A**) cf. *Coelodus* sp. (MTM V 2010.139.1). (**B**) Pycnodontiformes incertae sedis (UFRJ-DG 423-PD). (**C**) *Agassizilia erfoudina* (FSAC-KK 5073). (**D**) *Neoproscinetes africanus* (FSAC-KK 5070). (**E**) *Turbomesodon* cf. *arcuatus* (NHM PV P.45763). Collection abbreviations: FSAC—Faculté des Sciences Aïn Chock; MTM—Hungarian Natural History Museum (Magyar Természettudományi Múzeum); NHM—Natural History Museum, London; UFRJ—Universidade Federal do Rio de Janeiro. Figures of *Turbomesodon* cf. *arcuatus*, *Neoproscinetes africanus*, *Agassizilia erfoudinia*, Pycnodontiformes incertae sedis and *Coelodus* sp. modified from [29,47,151,200] respectively. Scale bar for (**A**,**C**–**E**) is 10 mm. Scale bar for (**B**) is 5 mm.

Sedimentation of the Csehbánya Fm. was determined to have been a floodplain environment [201]. Work on the pycnodont dentitions from multiple authors has revealed quite a lot about the paleoecology of these pycnodonts. Stable isotope composition performed on their teeth revealed low concentrations of strontium and that migration from marine or brackish environments were thus unlikely [197]. Additionally, different ontogenetic stages are represented, evidencing that these pycnodonts inhabited these non-marine environments for their entire life [199]. Botfalvai et al. [198] revealed that these pycnodonts were comparatively abundant in the vertebrate assemblage, being the third most frequent group represented by 28 individuals. An abundance of potential molluscan prey, such as bivalves and gastropods, are known from the Csehbánya Formation, and Szabó et al. [199] hypothe-

sized that these would have formed the main food source of the pycnodonts. However, so far, no direct evidence of a predator-prey relationship between these pycnodonts and mollusks has been observed.

Another Hungarian pycnodont represented by isolated molariform oral and hook-like branchial teeth, which is currently assigned to Pycnodontiformes indet., was recovered from the Santonian Ajka Coal Fm. [202]. This pycnodont also can be determined to be a freshwater inhabitant as the Ajka Formation was dominated by swampy and lacustrine environments in multiple sub basins [203–205].

### 2.4.5. France

Fragmentary vomerine and prearticular dentitions identified as cf. *Phacodus* were recovered from the Campanian Villeveyrac—L'Olivet locality in southern France [45]. The fossil assemblage of this site, particularly the remains of plants and mollusks such as gastropods and bivalves, correspond to taxa that are typically known from lacustrine environments [206]. As with cf. *Anomoeodus* from the Barremian Sao Khua Fm. of Thailand, Cavin et al. [45] also observed that all specimens of cf. *Phacodus* from this locality were small (e.g., the vomer was estimated to be roughly 5 mm long if complete). Using the proportions of a typical pycnodont [63], it then corresponds to a total length of just over 5 cm. All other known records of *Phacodus* show it to be a marine fish with specimens 20 times larger than the vomer from Villeveyrac—L'Olivet. Assuming that the French specimens represent juveniles, *Phacodus* might have represented an anadromous fish, with adults returning to freshwater to spawn and juveniles traveling to marine waters once they reached adulthood.

### 2.4.6. Spain

Isolated teeth belonging to unidentified pycnodonts were reported by many authors in the Campanian–Maastrichtian locality of Lo Hueco of the Province of Cuenca in eastern Spain [207–210], which was a swamp or wetland type environment based on palyno-logical analysis [209]. During the Maastrichtian, a number of freshwater and estuarine environments from northeastern Iberia [211] were home to several taxa of pycnodonts but unfortunately, all of these are represented by isolated teeth only and are thus rarely assigned to the genus level. These environments include the brackish coastal plain Fontllonga-6 locality (Pycnodontiformes indet., ?Pycnodontiformes indet.); the coastal lagoonal site of Els Nerets (cf. *Coelodus* sp.), which is interpreted to be predominantly freshwater due to the richness of cypriniforms and freshwater zygospores in the lower beds from the site but also to contain some marine influence due to the presence of dinoflagellate cysts in the upper beds; the estuarine wetland L'Espinau site (?Pycnodontiformes indet., cf. *Coelodus* sp.); the floodplains of Serrat del Rostiar-1 (Pycnodontiformes indet.), which contain ephemeral ponds and fluvial channels with some marine influence; and the riverine environment of the Camí del Soldat locality of the Tremp Fm. (Pycnodontiformes indet.), which had stable salinity due to tidal currents accumulating seawater in shallow pools at the margins of the river [212].

### 2.4.7. North America

Jaws and teeth belonging to two indeterminate species of pycnodontids were uncovered from the Mussentuchit Member of the Cenomanian Cedar Mountain Fm. in Emery County, Utah [213,214], which represents a broad alluvial floodplain with no evidence of marine or brackish water facies [215]. The Cenomanian Dakota Fm. of the Grand Staircase–Escalante National Monument region of Utah, which was a floodplain environment, yielded vomerine and prearticular tooth plates, branchial teeth, and contour scales that were assigned to *Coelodus* sp. by Brinkman et al. [216]. Fragmentary vomers, isolated vomerine teeth, branchial teeth, and contour scales belonging to *Coelodus* sp. were also uncovered from the Turonian Smoky Hollow Member of the Straight Cliffs Formation in Utah, which contains sediments, indicating a lacustrine/floodplain paleoenvironment from

the middle Turonian [216]. Additionally, isolated vomerine and branchial teeth of *Coelodus* sp. were found in the Coniacian John Henry Member of the Straight Cliffs Formation. These Coniacian specimens are also considered to be non-marine as they co-occur with teeth of the freshwater shark *Lonchidion* [217,218]. Brinkman et al. [216] also reported isolated oral and branchial teeth of *Micropycnodon* sp. from two brackish localities, the Santonian Straight Cliffs and Campanian Wahweap formations, with both localities also being in southern Utah.

2.4.8. South America

Just as in the Early Cretaceous, South America provided a wide assortment of non-marine Late Cretaceous pycnodonts. The most diverse continental vertebrate fauna of Late Cretaceous age of South America comes from the Cenomanian Alcântara Formation of Brazil [219,220]. Here, the Laje do Coringa outcrop is the richest fossil site of the Alcântara Fm. and shows high taxonomic similarities with the vertebrate fauna of the Kem Kem beds, which is related to South America and Africa drifting apart in the late Aptian–Albian [221]. Pycnodonts of the Laje do Coringa assemblage are represented by isolated remains of a single indeterminate pycnodontiform [222]. The paleoenvironment of Laje do Coringa was inferred by Medeiros et al. [221] as an estuary located in a forested coastline in a semi-arid-to-arid climate.

In Argentina, isolated vomerine and prearticular dentitions of at least two pycnodontiforms were recovered from non-marine sediments of the Maastrichtian Yacoraite Fm. Several dental remains identified as *Coelodus toncoensis* were reported from gray and black shale levels of the Amblayo Member in the Salta Basin [223–227]. Sedimentation of the Amblayo Member was predominantly marine but becomes increasingly mixed with freshwater and decreases in sea level near the boundary with the overlying Güemes Member, as can be seen in the low values of oxygen and carbon isotopes obtained from the deposited limestone [228]. This environmental change can also be observed in the increase in terrigenous content and the accumulation of silty red sandstone and mudstone–wackestone facies [228]. Both types of evidence indicate that *C. toncoensis* occupied a shallow-water environment with varying salinity. The gray and black shales from which the remains of *C. toncoensis* were recovered are indicators of a profundal setting within the center of a lacustrine basin [229].

Additionally, an indeterminate pycnodontiform was reported from the Tres Cruces sub-basin of the Yacoraite Fm. [230]. Marquillas et al. [228] conducted an isotope analysis of the limestones of this sub-basin and found a similar decrease in oxygen and carbon isotope values as was found for the upper Amablayo Member, suggesting a similar brackish environment for this pycnodont as for *C. toncoensis*.

The only other South American country where freshwater pycnodonts were recorded is Bolivia. Here, a tooth of an indetermined pycnodont was described from the Campanian Chaunaca Fm., representing a continental fluvial deposit [231,232]. Three pycnodont taxa were recovered from the brackish/lagoonal Paja Patcha locality of the Lower El Morino Formation, which is Maastrichtian in age: branchial teeth assigned to an indeterminate pycnodontid [233,234]; a fragmentary prearticular of *Coelodus* sp. [234]; and fragmentary prearticulars, vomers, and isolated teeth belonging to *Coelodus toncoensis* [233–235]. An indeterminate pycnodontid also was reported by de Muizon et al. [236] from the marshy to palustral Vila Vascarra locality, which also belongs to the El Morino Fm.

2.4.9. Africa

Taxonomically undiagnostic pycnodont teeth (Pycnodontiformes indet.) from the Cenomanian "Continental Intercalaire" of the Guir Basin of Algeria were reported by Benyoucef et al. [237]. One tooth is more rounded than the other and unornamented, while the other tooth is ornamented by a series of crenulations surrounding an apical depression. They were recovered from the Menaguir site, which is interpreted to be a

nearshore environment based on sedimentological and ichnological features implying brackish water conditions [237].

Two taxa of Maastrichtian non-marine pycnodonts, *Pycnodus* sp. and *Pycnodus jonesae*, were reported from the Ménaka Formation in the Iullemmeden Basin, Mali [238,239]. Both *Pycnodus* taxa were deposited in a phosphate conglomerate, which is indicative of shallow-marine-to-brackish environmental conditions rather than freshwater conditions [240]. *Pycnodus* sp. is represented by fragmentary tooth plates, which are insufficiently preserved to assign to a species level. Specimens of *Pycnodus jonesae* are represented by isolated teeth, prearticular dentitions and indeterminate tooth plates. However, the assignment of these remains to *Pycnodus* are dubious and they certainly represent other, probably new pycnodontiforms (compare [42]).

### 2.4.10. Madagascar

Only one possible freshwater pycnodont, represented by an incomplete vomerine tooth plate assigned to *Coelodus* sp., has been reported from Madagascar up to now, which is Maastrichtian in age [241]. This specimen comes from the Anembalemba Member of the Maevarano Formation in Boeny, which is interpreted as a coastal alluvial plain that was subject to strong seasonality with intense rainfall followed by periods of drought based on sedimentary features in the facies deposits, showcasing alternating episodes of turbulent flow and mass flow [242–244]. The Maevarano Formation, being coastal and being overlain by the Maastrichtian marine Berivotra Formation, could explain why *Coelodus* and other marine fish taxa were present in this continental setting as marine transgression could allow for easier access to such environments [242].

### *2.5. Paleogene*

Pycnodont diversity in the Paleogene drops significantly compared to what is seen in the prior Late Cretaceous [16,19], and as a result, less non-marine occurrences of pycnodonts were identified in the fossil record.

### 2.5.1. India

Paleogene non-marine pycnodontiforms of India were reported from the intertrappean Andhra Pradesh Formation at Rajahmundry, including two taxa, *Pycnodus* sp. and *Pycnodus* cf. *P. praecursor* [183]. These fossiliferous layers were deposited just after the K/Pg extinction event during the early Danian according to 40Ar/39Ar radioisotopic dating and paleomagnetism [245]. Branchial teeth originally identified as *Eotrigonodon* aff. *E. jonesi* and also found in Rajahmundry are now considered to belong to a pycnodontid [168,246]. Additional teeth of unidentified pycnodontids were additionally uncovered from this site [247]. All of these fishes were deposited in silty claystones, which reflect supratidal-to-terrestrial environments, followed by a return to restricted brackish conditions [248].

### 2.5.2. North America

Isolated molariform teeth of *Pycnodus* sp. have been collected from both the Tallahatta (Ypresian–Lutetian) and Lisbon formations (Lutetian) of the Claiborne Group in Alabama, USA [249]. The paleoenvironment of the Claiborne Group was most likely a marine environment with a soft/muddy bottom based on an analysis of otoliths of various teleosts [249]. However, the presence of teleost families in this locality known to occupy brackish waters [249], as well as fossils of turtle taxa that occur in estuaries today being found in the same locality [250], indicate that this pycnodont might have been an inhabitant of an estuarine environment.

### 2.5.3. South America

Benedetto and Sanchez [224] report *Coelodus toncoensis* not only from the Maastrichtian Amblayo Member of the Yacoraite Formation but also from the limestones of the Yacoraite Formation close to the boundary of the Santa Barbara Formation. This description

correlates well with the Alemanía Member of the Yacoraite Formation, which is of Danian age [228,229], indicating that this species might have survived the K-Pg extinction event. In the Paleogene part of the Yacoraite Fm., a return to more marine conditions can be observed based on a change from negative to positive values in carbon and oxygen isotopes [228]. However, paleontological evidence such as the presence of euryhaline invertebrates, including ostracodes, bivalves, and gastropods, indicate a shallow marine environment with some localized mixing of fresh and brackish waters [251], which matches the mixed environment of the upper Amblayo Member (Maastrichtian) of the Yacoraite Fm., where *C. toncoensis* was also present.

### 2.5.4. Africa

Like with the *Pycnodus* specimens from the Maastrichtian, the Paleogene pycnodonts from Mali were also found in phosphatic conglomerates, indicating shallow-marine-to-brackish conditions. An earlier Malian representative of *Pycnodus* from the Paleocene (Selandian–Thanetian), *P. jonesae*, might have survived the K-Pg extinction event and was recovered from the Teberemt Formation of the Gao Trench and Iullemmeden basins. Additionally, *P. jonesae* was found in the late Paleocene (Thanetian) Teberemt Fm. of the Iullemeden Basin [238,239]. *Pycnodus* sp. was also reported from the Teberemt Fm. The most famous of these localities is located in the Tamaguélelt Fm. of the Taoudenit Basin, Mali, which yielded three taxa—*Pycnodus*, *Pycnodus* sp., *P. zeaformis*, and *P. maliensis*—which are all Eocene (Ypresian) in age [238,239]. It is noteworthy that *P. maliensis* is represented by prearticular remains, while *P. zeaformis* is known by vomers, indicating that both could be synonymous.

### 3. Discussion

One clear pattern arises when the entire non-marine pycnodont fossil record is considered: the incredible rise in such occurrences throughout the Cretaceous. While Jurassic occurrences are sporadic, with low generic diversity, this changes dramatically when one enters the Early Cretaceous, with 14 known genera observed during this time period, and the number of occurrences is eleven times higher than the preceding period (Tables 1 and 2). A map of these occurrences shows that pycnodonts were initially present in far eastern Asia, North America, and central Africa during the Late Jurassic and extended their range into western Europe, South America, and northern Africa in the Early Cretaceous (Figure 3). The number of occurrences and known genera both decrease in the Late Cretaceous, with the latter being especially affected by a poorer fossil record with six known genera. Late Cretaceous non-marine occurrences also see the first appearance of pycnodonts in both Madagascar and India. There is a westward shift throughout the late Cretaceous in the distribution of non-marine South American pycnodonts from Brazil in the Cenomanian to Bolivia and Argentina in the Campanian–Maastrichtian.

**Table 1.** Non-marine pycnodont occurrences and diversity through time. Only specimens from localities with known stratigraphic ages are included.

| Time Bin | Non-Marine Occurrences | No. of Known Genera |
| --- | --- | --- |
| Late Jurassic | 3 | 1 |
| Early Cretaceous | 60 | 14 |
| Late Cretaceous | 45 | 5 |
| Paleocene | 7 | 2 |
| Eocene | 5 | 1 |

**Table 2.** Non-marine pycnodont occurrences and diversity through time. Localities with uncertain stratigraphic age included.

| Time Bin | Non-Marine Occurrences | No. of Known Genera |
| --- | --- | --- |
| Late Jurassic | 4 | 3 |
| Early Cretaceous | 61 | 15 |
| Late Cretaceous | 50 | 8 |
| Paleocene | 9 | 2 |
| Eocene | 5 | 1 |

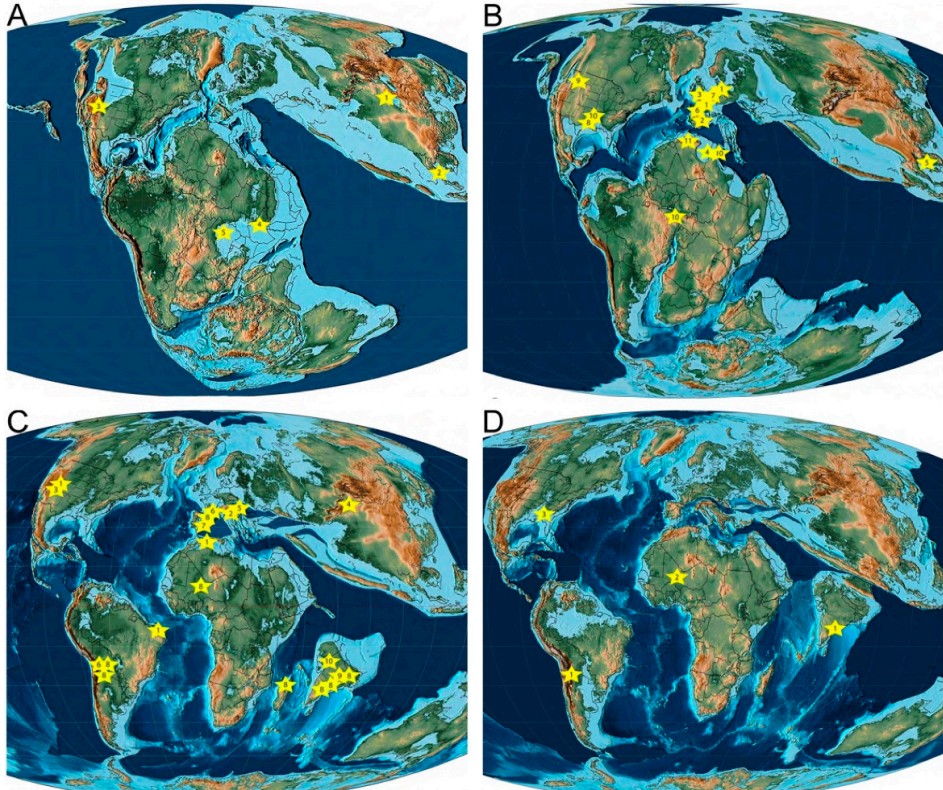

**Figure 3.** Paleobiogeographic distribution of non-marine pycnodonts through time. Numbered stars refer primarily to geological age. When geological age is unclear, the star will represent the locality the fossil was collected from as well as the currently time span that the locality is believed to represent. All cited localities and localities referring to a particular stage are discussed in further detail within the main text. Maps modified from Scotese [252–255]. (**A**) Late Jurassic, Oxfordian, 158.4 Ma: 1. Changtu, Tibet, Late Jurassic; 2. Khlong Min Formation, Thailand, Mid–Late Jurassic; 3. Morrison Formation, USA, Oxfordian–Tithonian; 4. Mugher Mudstone, Ethiopia, ?Kimmeridgian–Tithonian; 5. Stanleyville Formation, Democratic Republic of Congo, Kimmeridgian–Valanginian. (**B**) Early Cretaceous, early Aptian, 121.8 Ma: 1. Berriasian; 2. Iberian Basin, eastern Spain, Berriasian–Valanginian, Barremian–Albian; 3. Southern England, Berriasian–Valanginian, Hauterivian–Barremian; 4. Valanginian 5. Phu Phan Thong, Thailand, Hauterivian–Barremian; 6. Vega de Pas Formation, northern Spain, Hauterivian–Barremian; 7. Bernissart, Belgium, Barremian–Aptian; 8. Paluxy Formation, northern Texas, USA, Albian–Aptian; 9. Aptian; 10. Albian; 11. Kem Kem, Albian–Cenomanian. (**C**) Late Cretaceous, Maastrichtian, 68 Ma: 1. Cenomanian; 2. Turonian; 3. Southern Utah, USA, Turonian–Campanian; 4. Donggou Formation, China, Coniacian–Campanian; 5. Santonian; 6. Campanian; 7. Lo Hueco (Cuenca), Spain, Campanian–Maastrichtian; 8. Maastrichtian; 9. Jhilmili, India, Maastrichtian–Danian; 10. Rajasthan, India, Maastrichtian–Selandian. (**D**) Paleogene, Ypresian, 52.2 Ma: 1. Danian; 2. Teberemt and Tamaguélelt Formations, Mali, Selandian–Ypresian; 3. Lutetian.

Aside from taxa that are difficult to assign taxonomically, only two families of pycnodontiform fishes are known to enter non-marine habitats: Mesturidae and Pycnodontidae, the latter which accounts for the vast majority of such environmental transitions. When seen as a time-calibrated phylogeny, it can be observed that multiple transitions have been made throughout the order of Pycnodontiformes, with just the two species of pycnodontids *Turbomesodon praeclarus* and *T. bernissartensis* sharing a common ancestor (Figure 4). Given that the fossil record of non-marine pycnodonts is so fragmentary, relationships between many of the taxa in this phylogeny are poorly resolved, and more complete future finds will be necessary before reconstructing the frequency of marine-to-brackish-to-freshwater transitions within this group.

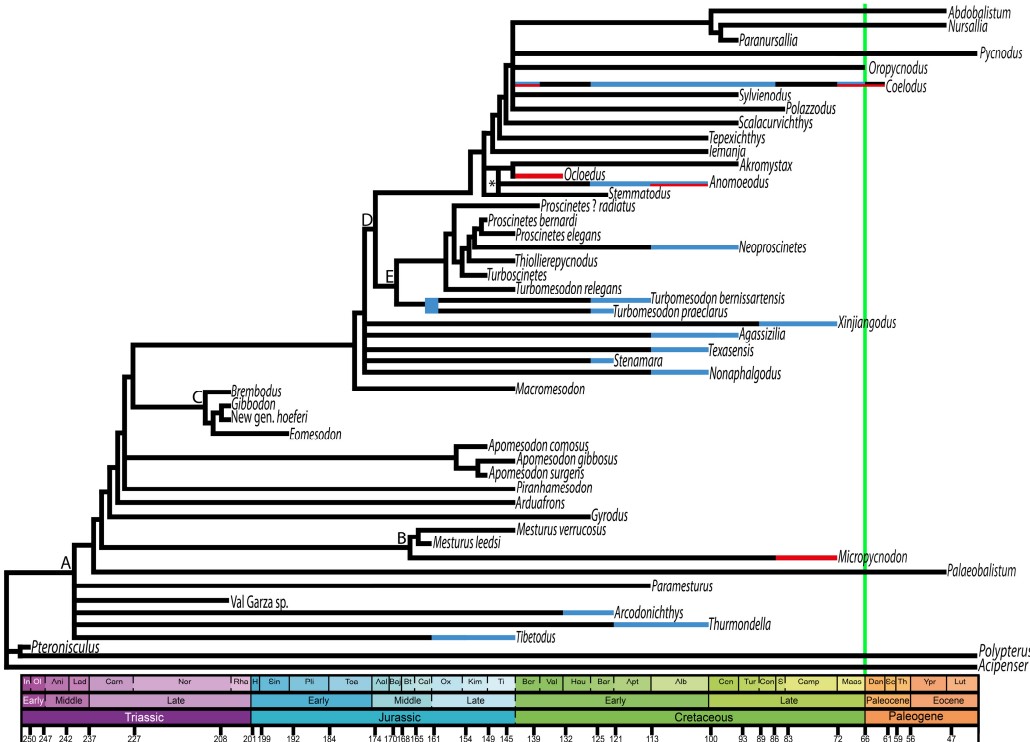

**Figure 4.** Distribution of non-marine habitat transitions throughout a time-calibrated phylogeny of Pycnodontiformes based on work by Ebert [124]. The green vertical line represents the K/Pg boundary. Freshwater and brackish environments are represented by blue and red bars, respectively. Branches which contain both red and blue bars indicate the occupation of both environments by that taxon during that particular geological stage. Uppercase letters at certain nodes represent monophyletic groups: A: Pycnodontiformes; B: Mesturidae; C: Brembodontidae; D: Pycnodontidae; E: Proscinetinae; * represents indeterminate position of *Anomoeodus* within Pycnodontidae based on a previous phylogenetic analysis [48]. Geological ages are represented by abbreviations which are listed here in chronological order: In, Induan; Ol, Olenkian; Ani, Anisian; Lad, Ladinian; Carn, Carnian; Nor, Norian; Rha, Rhaetian; H, Hettangian; Sin, Sinemurian; Pli, Pliensbachian; Toa, Toarcian; Aal, Aalenian; Baj, Bajocian; Bt, Bathonian; Cal, Callovian; Ox, Oxfordian; Kim, Kimmeridgian; Ti, Tithonian; Ber, Berriasian; Val, Valanginian; Hau, Hauterivian; Bar, Barremian; Apt; Aptian; Alb, Albian; Cen; Cenomanian; Tur, Turonian; Con, Coniacian; S, Santonian; Camp, Campanian; Maas, Maastrichtian; Dan, Danian; Se, Selandian; Th, Thanetian; Ypr, Ypresian; Lut, Lutetian. Figure produced with TSCreator (https://timescalecreator.org/, accessed 19 March 2024).

It is only in the Cretaceous that fully articulated fossil non-marine specimens are found, such as *Stenamara mia* and *Turbomesodon praeclarus* [48,76] from the Barremian Las Hoyas Fm. in Spain and *Pycnodus lametae* [167,168,173] from the Maastrichtian Lameta Fm. in India, which has greatly helped researchers in the accurate classification of specimens, whereas the Jurassic and Paleogene non-marine pycnodonts are represented by far more

fragmentary material. Also of note is that all articulated pycnodont specimens are found in lacustrine deposits but are a significant minority of non-marine pycnodont specimens discovered so far (5 out of 129 listed occurrences, or 3.88%) (see Appendix A). Although the preservation quality of the freshwater pycnodont fossil record is generally poor, with the few exceptions mentioned above, Guinot and Cavin [256] detected a major decrease in body size among pycnodonts when they moved into freshwater habitats.

The largest numbers of Cretaceous non-marine pycnodonts occurred during the earliest (Berriasian) and latest (Maastrichtian) Cretaceous, with 19 and 29 taxa, respectively (Figure 5, Table 3). Pycnodonts being so common in non-marine habitats during the Cretaceous could be a result of the environmental conditions throughout much of this time. The Early Cretaceous, in particular, was a time of immense environmental change, with not only sea level changes but a shift from an arid to a humid climate [257]. Sea levels were much higher than today due to the absence of polar ice caps. Such variations in sea level increase the possibility of marine fishes colonizing continental environments and were a major factor explaining why many marine fish lineages made incursions into freshwater environments during the Miocene [258]. Low sea levels, conversely, increased the spread of low-lying areas, which would have allowed for the movement of freshwater fishes from one hydrographic basin to another, and which could thus have been an additional factor for why freshwater occurrences in pycnodonts rose so dramatically from the Late Jurassic to the Early Cretaceous (Tables 1 and 2, Appendix A).

**Table 3.** Non-marine pycnodont occurrences during each stage in the Cretaceous in total numbers of occurrences and in relation to total non-marine pycnodontiform occurrences throughout the Cretaceous.

| Stage | Non-Marine Occurrences | % of Cretaceous Non-Marine Occurrences |
|---|---|---|
| Berriasian | 19 | 17.3 |
| Berriasian–Valanginian | 2 | 1.8 |
| Valanginian | 2 | 1.8 |
| Hauterivian–Barremian | 3 | 2.7 |
| Berriasian–Barremian | 1 | 0.9 |
| Barremian | 13 | 11.8 |
| Barremian–Aptian | 1 | 0.9 |
| Aptian | 8 | 7.3 |
| Albian | 11 | 10.0 |
| Albian–Cenomanian | 5 | 4.5 |
| Cenomanian | 5 | 4.5 |
| Turonian | 2 | 1.8 |
| Coniacian | 1 | 0.9 |
| Coniacian–Campanian | 1 | 0.9 |
| Santonian | 3 | 2.7 |
| Campanian | 3 | 2.7 |
| Campanian–Maastrichtian | 1 | 0.9 |
| Maastrichtian | 29 | 26.4 |

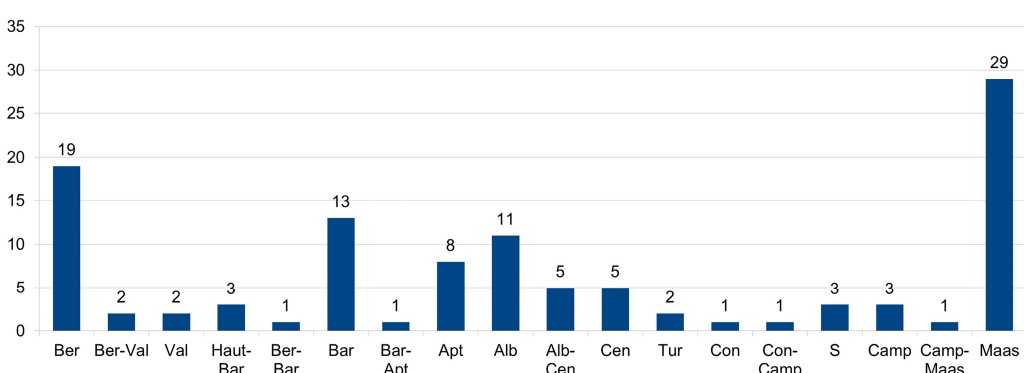

**Figure 5.** Total number of non-marine pycnodont occurrences from each geological stage of the Cretaceous.

Epicontinental seas were a common phenomenon during the Cretaceous, such as the Western Interior Seaway (Albian–Maastrichtian [259,260]) and the Trans-Saharan Seaway (Cenomanian–Ypresian [239]). Due to the increase in sea levels, many brackish water ecosystems and interior seas developed and would have encountered extreme variations in salinity. This probably explains why many fossil localities from the Cretaceous contain mixed assemblages of both freshwater and marine fish taxa, making habitat interpretations based on fish taxa alone difficult and prone to errors. There was also a period of high sea levels during the Paleogene, such as the Paleocene–Eocene Thermal Maximum event (56 MYA), which coincides with the exclusive presence of pycnodonts in brackish water environments (Appendix A), e.g., those occurring in Mali [239]. The increase in freshwater habitat occupation during the Cretaceous may reflect a genuine signal, as similar movements into these habitats can be observed in other fish groups at the time, such as Ginglymodi [13] and Hybodontiformes [11].

Freshwater habitats being used as a refugia for ancient fish clades is a consistent evolutionary pattern and was noted in Amiiformes, Lepisosteiformes, and Dipnoi [11,261–263]. Cavin [10] stated that freshwater environments appeared to be far less vulnerable to environmental disturbance than marine habitats during the K-Pg extinction event. The majority of fish clades (14 of 15 families) that became extinct at this time were marine forms that were also piscivores at the top of the food chain [264], and no exclusively freshwater clade was wiped out during the K-Pg boundary event. An analysis by Sheehan and Fastovsky [265] revealed that over 80% of the actinopterygian species from freshwater deposits in eastern Montana crossed the K-Pg boundary. Similarly, work by Cavin and Martin [266] showed that most actinopterygian families (24 out of 25) crossing the K-Pg boundary had representatives living in freshwater or brackish environments, ranging from those living exclusively in freshwater to marine groups that had occasional freshwater/brackish representatives. The reason for this discrepancy in victims of this mass extinction most likely lies in how their food webs were structured. Marine food chains, particularly those of the open ocean, have phytoplankton at the base of the food chain, which was drastically affected by the darkening of the atmosphere from the asteroid impact, and which ultimately had a domino effect on the rest of the ecosystems until it hit the apex predators [267]. This is supported by the documented decline in pelagic plankton after the K-Pg boundary [268], which agrees well with the disappearance of pelagic predatory fish.

The food webs of freshwater ecosystems, in contrast, rely on detritus, which continues to function regardless of the potential for photosynthesis being diminished. Supporting this further, the two families of fishes that were not strictly marine that perished during the K-Pg extinction event had representatives that would have been planktivorous, e.g., the mawsoniid sarcopterygians and aspidorhynchid actinopterygians [10,269]. Coastal marine and brackish habitats would also be able to weather the extinction event better as they also had access to the detritus from river systems that flowed into the ocean [267]. Since pycnodonts not only occupied numerous freshwater ecosystems at the boundary but were predominantly present in coastal habitats such as reefs, this could be one particular factor

explaining why they survived the K-Pg extinction event while many other marine fish clades were wiped out. Two genera in particular, *Coelodus* and *Pycnodus*, were present in freshwater before the K/Pg boundary and predominantly occupied brackish ecosystems, which might have acted as refugial environments for these two taxa, permitting them to survive the end-Cretaceous extinction event (Figure 4). *Abdobalistum*, *Nursallia*, and *Paleobalistum*, however, survived the K-Pg extinction event in marine environments. *Oropycnodus*, also occupying marine environments, is seemingly the only pycnodont that went extinct during the end-Cretaceous extinction event. Consequently, freshwater environments might have only played a minor role as refugium for pycnodonts in their survival of the K-Pg extinction event.

## 4. Conclusions

Pycnodonts were a predominantly marine group that were especially prevalent in structured habitats such as reefs from the Norian (Late Triassic) to the Ypresian (middle Eocene, Mid-Paleogene), spanning an evolutionary history of ca. 170 Ma [42]. Nevertheless, they seemingly made numerous incursions into brackish and even freshwater habitats throughout their evolutionary history, starting in the Jurassic but particularly during the Cretaceous. The last freshwater occurrences coincide with those from marine deposits [42,249]. The occurrence in the Lutetian Lisbon Formation of the Claiborne Group in Alabama, USA [249], could even indicate that pycnodonts survived longer in freshwater than in seawater if the stratigraphic assignment is correct. However, despite their repeated incursion into continental environments, pycnodontiforms seemingly never became permanent inhabitants of freshwater environments like more ancient clades, e.g., Lepisosteiformes and Amiiformes.

The marine-to-freshwater transition of pycnodonts appeared to be more similar to that seen in younger marine-derived fish clades today, such as silversides, drums, needlefishes, anchovies, and pufferfishes [270–272], whereby a mostly marine clade makes numerous incursions into freshwater habitats throughout their evolution but has the bulk of their diversity present in marine habitats. The freshwater fossil record of pycnodontiforms is, for the most part, very fragmentary, represented by isolated dental remains. Only a few taxa from the Cretaceous are preserved as fully articulated specimens. This is an issue for most freshwater fish fossils as certain habitats, particularly fluvial, are hostile to fossilization due to the high-energy movements of currents. However, despite these limitations, there are certain clues about ecology and life history that can be deduced from certain taxa and their occurrences, such as pycnodontiforms from the Early Cretaceous of Spain [120] and Hungary [197,199] being permanent freshwater residents due to the wide range of body and tooth plate sizes, respectively; their being present in the same locality, indicating the simultaneous presence of juveniles and adults, whereas other pycnodont taxa migrated to freshwaters to reproduce, while otherwise living in marine environments (as seen in cf. *Phacodus* [45]); and finally, the possibility of niche partitioning, as seen in the Kem Kem pycnodont assemblage [47].

Our review of the freshwater pycnodont fossil record is far from being complete; in fact, we assume it to be rather incomplete due to the taphonomic processes and insufficient data. But even with such limited data, fascinating and tantalizing glimpses have already been obtained from these specimens. A renewed collection effort in freshwater localities, alongside the discovery of new fossil sites combined with isotopic analyses, should assist immensely in not only further discovering the extent of pycnodont diversity in non-marine environs but, hopefully, in shedding further light on evolutionary questions concerning both extinct and extant clades of freshwater fishes and how they came to dominate the rivers and lakes of the world.

**Author Contributions:** Conceptualization, J.J.C. and J.K.; writing—original draft preparation, J.J.C.; writing—review and editing, J.K. All authors have read and agreed to the published version of the manuscript.

**Funding:** Open Access Funding by the University of Vienna.

**Data Availability Statement:** All data used for this review are provided in the main body of the text.

**Acknowledgments:** This work is the result of many years of research into pycnodontiform fishes. Many people contributed to the various studies by providing access to collections under their care. It is impossible to name all of them and we apologize to everybody who is missing in the following list: G. Arratia and H.-P. Schultze (formerly Natural History Museum Berlin, Germany, now Biodiversität Institute and Natural History Museum, The University of Kansas); G. Bergér (Museum Bergér, Eichstätt, Germany); R. Böttcher (now retired) and E. Maxwell (Staatliches Museum für Naturkunde, Stuttgart, Germany); the late P. Forey, A. Longbottom (now retired), Z. Johanson, and E. Bernard (The Natural History Museum London, UK); M.A. Herrero (Museo Paleontológico de Galve, Spain); M. Kölbl-Ebert and C. Ifrim (Jura-Museum Eichstätt, Germany); the late M. Mäuser and O. Wings (Naturkunde Museum Bamberg, Germany); W. Munk (Staatliches Museum für Naturkunde Karlsruhe, Germany); O.W.M. Rauhut, M. Kölbl, W. Werner (now retired), and M. Krings (all Bayerische Staatssammlung für Geologie und Paläontologie), who are all acknowledged for supporting our research. Finally, we would like to thank the three reviewers of the present work for their comments and valuable insights, which helped improving the manuscript. Open access funding was provided by the Library of the University of Vienna.

**Conflicts of Interest:** The authors declare no conflicts of interest.

## Appendix A

Taxonomy, affinities, and paleogeographic distribution of non-marine pycnodontiforms in stratigraphic order from oldest to youngest. *, may be *Ocloedus*; **, occurred in brackish habitat; ***, lost during preparation.

| Taxa | Family | Country | Locality | Period | Stage | Ref. |
|------|--------|---------|----------|--------|-------|------|
| cf. *Gyrodus* | Gyrodontidae | Thailand | Khlong Min Formation (Mab Ching) | Middle–Late Jurassic | Unknown | [21] |
| *Tibetodus gyroides* | Family incertae sedis | China | Tibet | Late Jurassic | Unknown | [25,26] |
| Pycnodontoidea gen et spp. indet | Family incertae sedis | USA | Morrison Formation, Dinosaur National Monument | Late Jurassic | Oxfordian–Tithonian | [27] |
| cf. *Pycnodus* | Pycnodontidae | Ethiopia | Abay Gorge, Mugher Mudstone | Late Jurassic | ?Kimmeridgian–Tithonian | [39] |
| *Congopycnodus cornutus* | Family incertae sedis | Democratic Republic of Congo | Stanleyville | Late Jurassic-Early Cretaceous | Kimmeridgian–Valanginian | [32] |
| ?Pycnodontiformes indet. (tooth morphotype 2) | Family incertae sedis | France | Champblanc Quarry, Cherves-de-Cognac | Early Cretaceous | Berriasian | [77] |
| aff. *Gyrodus* (tooth morphotype 6) | Gyrodontidae | France | Champblanc Quarry, Cherves-de-Cognac | Early Cretaceous | Berriasian | [77] |
| cf. *Arcodonichthys* (tooth morphotype 7) | Family incertae sedis | France | Champblanc Quarry, Cherves-de-Cognac | Early Cretaceous | Berriasian | [77,80] |
| Pycnodontiformes indet. (tooth morphotype 8) | Family incertae sedis | France | Champblanc Quarry, Cherves-de-Cognac | Early Cretaceous | Berriasian | [77] |
| Pycnodontidae gen. et spp. Indet (tooth morphotype 9) | Pycnodontidae | France | Champblanc Quarry, Cherves-de-Cognac | Early Cretaceous | Berriasian | [77] |

| Taxa | Family | Country | Locality | Period | Stage | Ref. |
|---|---|---|---|---|---|---|
| Pycnodontidae gen. et spp. Indet (tooth morphotype 10) | Pycnodontidae | France | Champblanc Quarry, Cherves-de-Cognac | Early Cretaceous | Berriasian | [77] |
| Pycnodontidae gen. et spp. Indet (tooth morphotype 11) | Family incertae sedis | France | Champblanc Quarry, Cherves-de-Cognac | Early Cretaceous | Berriasian | [77] |
| *Coelodus* | Pycnodontidae | England | Lulworth Formation | Early Cretaceous | Berriasian | [26,49] |
| *Coelodus* ** | Pycnodontidae | England | Durlston Formation | Early Cretaceous | Berriasian | [26,49] |
| "*Pycnodus*" *mantelli* | Pycnodontidae | Germany | Gronau, North Rhine-Westphalia | Early Cretaceous | Berriasian | [29,60] |
| "*Pycnodus*" *mantelli* | Pycnodontidae | Germany | Osterwald/Otternhagen, Lower Saxony | Early Cretaceous | Berriasian | [29,58] |
| *Turbomesodon* cf. *arcuatus* | Pycnodontidae | Germany | Bückeberg Group, Istenberg Formation | Early Cretaceous | Berriasian | [29,63] |
| *Turbomesodon* cf. *arcuatus* | Pycnodontidae | Germany | North Rhine-Westphalia | Early Cretaceous | Berriasian | [29,60–62] |
| Pycnodontiformes indet. | Family incertae sedis | Germany | Bückeberg Group, Fuhse Formation | Early Cretaceous | Berriasian | [29,55,56] |
| Pycnodontiformes indet. | Family incertae sedis | Germany | Bückeberg Group, Istenberg Formation | Early Cretaceous | Berriasian | [29,57] |
| Pycnodontiformes indet. | Family incertae sedis | Germany | Bückeberg Group, Oesede Formation or Istenberg Formation | Early Cretaceous | Berriasian | [29,58] |
| Pycnodontiformes indet. | Family incertae sedis | Germany | Lobber Ort, Island of Rügen | Early Cretaceous | Berriasian | [29,59] |
| *Turbomesodon arcuatus* | Pycnodontidae | England | Middle Purbeck Limestone | Early Cretaceous | Berriasian | [29] |
| *Proscinetes* sp. cf. *P. bernardi* | Pycnodontidae | Spain | Lérida | Early Cretaceous | Berriasian | [93,129–132] |
| *Ocloedus subdiscus* ** | Pycnodontidae | Spain | El Montsec, Lérida | Early Cretaceous | Berriasian–Valanginian | [48,89,101] |
| *Turbomesodon laevidens* | Pycnodontidae | England | Middle Purbeck Limestone | Early Cretaceous | Berriasian–Valanginian | [29] |
| *Ocloedus* | Pycnodontidae | England | Pevensey Pit, Ashdown Brickworks quarry (Wealden Group) | Early Cretaceous | Valanginian | [50,51,53] |
| *Turbomesodon microdon* | Pycnodontidae | England | Grinstead Clay Formation | Early Cretaceous | Valanginian | [29] |
| cf. *Anomoeodus* | Pycnodontidae | Thailand | Sao Khua Formation (Phu Phan Thong) | Early Cretaceous | Hauterivian–Barremian | [21] |
| *Turbomesodon multidens* | Pycnodontidae | England | Weald Clay Group, Sevenoaks, Kent | Early Cretaceous | Hauterivian–Barremian | [29] |
| *Arcodonichthys pasiegae* | Family incertae sedis | Spain | Vega de Pas Formation, Basque-Cantabrian Basin | Early Cretaceous | Hauterivian–Barremian | [80] |
| Pycnodontiformes indet. | Family incertae sedis | Spain | El Castellamar Formation | Early Cretaceous | Berriasian–Barremian | [107] |

| Taxa | Family | Country | Locality | Period | Stage | Ref. |
|------|--------|---------|----------|--------|-------|------|
| *Stenamara mia* | Pycnodontidae | Spain | Las Hoyas | Early Cretaceous | Barremian | [118] |
| *Turbomesodon praeclarus* | Pycnodontidae | Spain | Las Hoyas | Early Cretaceous | Barremian | [76] |
| *Coelodus* * | Pycnodontidae | England | Wessex (Wealden Group) | Early Cretaceous | Barremian | [46] |
| Pycnodontiformes indet. | Family incertae sedis | England | Wessex (Wealden Group) | Early Cretaceous | Barremian | [46] |
| Pycnodontidae gen. et spp. indet | Pycnodontidae | Spain | Camarillas Formation | Early Cretaceous | Barremian | [109] |
| cf. *Proscinetes* | Pycnodontidae | Spain | Camarillas Formation | Early Cretaceous | Barremian | [109] |
| *Coelodus* | Pycnodontidae | Spain | Camarillas Formation | Early Cretaceous | Barremian | [109] |
| *Anomoeodus* | Pycnodontidae | Spain | Camarillas Formation | Early Cretaceous | Barremian | Kriwet, pers. obser. |
| Pycnodontiformes indet. | Family incertae sedis | Spain | Artoles Formation | Early Cretaceous | Barremian | Kriwet, pers. obser. |
| .cf. *Ocloedus* sp. 1 | Pycnodontidae | Spain | Huérgiuna Formation | Early Cretaceous | Barremian | [44] |
| cf. *Ocloedus* sp. 2 | Pycnodontidae | Spain | Huérgiuna Formation | Early Cretaceous | Barremian | [44] |
| *Anomoeodus nursalli* | Pycnodontidae | Spain | Huérgiuna Formation | Early Cretaceous | Barremian | [44] |
| *Coelodus* | Pycnodontidae | Spain | Aguilón | Early Cretaceous | Barremian | [93,128–131] |
| *Turbomesodon bernissartensis* | Pycnodontidae | Belgium | Bernissart | Early Cretaceous | Barremian–Aptian | [75,76] |
| "*Paleobalistum*" *geiseri* | Pycnodontidae | USA | Twin Mountains Formation, Paluxy Church | Early Cretaceous | Aptian | [140] |
| "*Proscinetes*" *texanus* | Pycnodontidae | USA | Twin Mountains Formation, Paluxy Church | Early Cretaceous | Aptian | [30,140] |
| *Coelodus* | Pycnodontidae | USA | Twin Mountains Formation, Paluxy Church | Early Cretaceous | Aptian | [140] |
| *Thurmondella estesi* | Family incertae sedis | USA | Twin Mountains Formation, Paluxy Church | Early Cretaceous | Aptian | [140,144] |
| Pycnodontidae indet. | Pycnodontidae | USA | Cloverly Formation, Little Sheep Mudstone Member | Early Cretaceous | Aptian | [132] |
| *Anomoeodus complanatus* | Pycnodontidae | Spain | Cintorres, Valencia | Early Cretaceous | Aptian | [93,128–131] |
| *Anomoeodus complanatus* | Pycnodontidae | Spain | Mirambel, Tereul | Early Cretaceous | Aptian | [93,128–131] |
| *Coelodus* sp. aff. *C. soleri* | Pycnodontidae | Spain | Morella, Valencia | Early Cretaceous | Aptian | [93,128–131] |
| aff. *Gyrodus* | Gyrodontidae | Tunisia | Jebel Boulouha North | Early Cretaceous | Albian | [151] |
| *Thurmondella estesi* | Family incertae sedis | USA | Paluxy Formation, Stephenville Printing | Early Cretaceous | Albian | [30,140] |

| Taxa | Family | Country | Locality | Period | Stage | Ref. |
|---|---|---|---|---|---|---|
| *Texasensis coronatus* | Pycnodontidae | USA | Paluxy Formation, Pecan Valley Estates | Early Cretaceous | Albian | [30,31,140] |
| *"Macromesodon" dumblei* | Family incertae sedis | USA | Paluxy Formation, Pecan Valley Estates | Early Cretaceous | Albian | [30,140] |
| *Nonaphalgodus trinitiensis* | Pycnodontidae | USA | Antlers Formation, Butler Farm | Early Cretaceous | Albian | [30,140] |
| *Anomoeodus caddoi* ** | Pycnodontidae | USA | Holly Creek Formation | Early Cretaceous | Albian | [136] |
| Pycnodontiformes indet. ** | Family incertae sedis | USA | Holly Creek Formation | Early Cretaceous | Albian | [136] |
| ?Pycnodontiformes indet. ** | Family incertae sedis | Tunisia | Oued el Khil | Early Cretaceous | Albian | [151] |
| Pycnodontiformes indet. | Family incertae sedis | Brazil | Açu Formation, Potiguar Basin | Early Cretaceous | Albian | [150] |
| *Anomoeodus* sp. aff. *A. muensteri* | Pycnodontidae | Spain | Condemios de Abajo, Castilla-La-Mancha | Early Cretaceous | Albian | [93,128–131] |
| Pycnodontiformes indet. | Family incertae sedis | Spain | Ceceda, Asturias | Early Cretaceous | Albian | [93,128–131] |
| *Neoproscinetes africanus* | Pycnodontidae | Morocco | Kem Kem | Early–Late Cretaceous | Albian–Cenomanian | [47] |
| cf. *Macromesodon* | Pycnodontidae | Morocco | Kem Kem | Early–Late Cretaceous | Albian–Cenomanian | [47] |
| *Agassizilia erfoudina* | ?Pycnodontidae | Morocco | Kem Kem | Early–Late Cretaceous | Albian–Cenomanian | [47] |
| cf. *Coelodus* | Pycnodontidae | Morocco | Kem Kem | Early–Late Cretaceous | Albian–Cenomanian | [47] |
| *Coelodus soleri* | Pycnodontidae | Spain | Girona, Catalonia | Early–Late Cretaceous | Albian–Cenomanian | [93,128–131] |
| Pycnodontiformes indet. ** | Family incertae sedis | Brazil | Alcântara Formation, Laje do Coringa | Late Cretaceous | Cenomanian | [220,222] |
| Pycnodontiformes indet. ** | Family incertae sedis | Algeria | Continental Intercalaire, Guir Basin | Late Cretaceous | Cenomanian | [237] |
| Pycnodontidae gen. et spp. Indet 1 | Pycnodontidae | USA | Cedar Mountain Formation | Late Cretaceous | Cenomanian | [213] |
| Pycnodontidae gen. et spp. Indet 2 | Pycnodontidae | USA | Cedar Mountain Formation | Late Cretaceous | Cenomanian | [213] |
| *Coelodus* | Pycnodontidae | USA | Dakota Formation | Late Cretaceous | Cenomanian | [216] |
| *Coelodus* | Pycnodontidae | USA | Straight Cliffs Formation, Smoky Hollow Member | Late Cretaceous | Turonian | [216] |
| Pycnodontiformes indet./Pycnodontidae indet. ** | ?Pycnodontidae | Austria | Gams, Schönleiten Formation | Late Cretaceous | Turonian | [191,193] |
| *Coelodus* | Pycnodontidae | USA | Straight Cliffs Formation, John Henry Member | Late Cretaceous | Coniacian | [216] |
| *Xinjiangodus gyrodoides* | Pycnodontidae | China | Donggou Formation, Junggar Basin | Late Cretaceous | Coniacian–Campanian | [162] |
| cf. *Coelodus* | Pycnodontidae | Hungary | Csehbánya Formation, Iharkút | Late Cretaceous | Santonian | [197,200] |

| Taxa | Family | Country | Locality | Period | Stage | Ref. |
|---|---|---|---|---|---|---|
| *Micropycnodon* ** | Family incertae sedis | USA | Straight Cliffs Formation, John Henry Member | Late Cretaceous | Santonian | [216] |
| Pycnodontiformes indet. | Family incertae sedis | Hungary | Ajka Formation | Late Cretaceous | Santonian | [202] |
| cf. *Phacodus* | Family incertae sedis | France | Villeveyrac—L'Olivet | Late Cretaceous | Campanian | [45] |
| *Micropycnodon* ** | Family incertae sedis | USA | Wahweap Formation | Late Cretaceous | Campanian | [216] |
| Pycnodontidae gen. et spp. indet | Pycnodontidae | Bolivia | Chaunaca Formation | Late Cretaceous | Campanian | [231] |
| Pycnodontoidea gen. et spp. indet | Family incertae sedis | Spain | Lo Hueco (Cuenca) | Late Cretaceous | Campanian–Maastrichtian | [207–210] |
| *Pycnodus lametae* | Pycnodontidae | India | Lameta Formation (Maharashtra) | Late Cretaceous | Maastrichtian | [167,168,173] |
| cf. *Coelodus* ** | Family incertae sedis | Spain | Els Neret | Late Cretaceous | Maastrichtian | [211] |
| cf. *Coelodus* ** | Family incertae sedis | Spain | L'Espinau | Late Cretaceous | Maastrichtian | [211] |
| Pycnodontiformes indet. ** | Family incertae sedis | Spain | Serrat del Rostiar-1 | Late Cretaceous | Maastrichtian | [211] |
| Pycnodontiformes indet. ** | Family incertae sedis | Spain | Cami del Soldat | Late Cretaceous | Maastrichtian | [211] |
| Pycnodontiformes indet. ** | Family incertae sedis | Spain | Fontllonga-6 | Late Cretaceous | Maastrichtian | [211] |
| ?Pycnodontiformes indet. ** | Family incertae sedis | Spain | Fontllonga-6 | Late Cretaceous | Maastrichtian | [211] |
| ?Pycnodontiformes indet. ** | Family incertae sedis | Spain | L'Espinau | Late Cretaceous | Maastrichtian | [211] |
| Pycnodontidae gen. et spp. indet | Pycnodontidae | India | Madhya Pradesh (Kisalpuri) | Late Cretaceous | Maastrichtian | [179] |
| Pycnodontidae gen. et spp. indet | Pycnodontidae | India | Lameta Formation (Pisdura) | Late Cretaceous | Maastrichtian | [168,174,175] |
| *Pycnodus bicresta* | Pycnodontidae | India | Naskal, Andhra Pradesh | Late Cretaceous | Maastrichtian | [180,181] |
| Pycnodontidae gen. et spp. Indet ** | Pycnodontidae | India | Chhindwara District, Madhya Pradesh | Late Cretaceous | Maastrichtian | [185] |
| *Pycnodus* | Pycnodontidae | India | Papro Formation, Uttar Pradesh | Late Cretaceous | Maastrichtian | [176] |
| *"Pycnodus" lametae* *** | Pycnodontidae | India | Naskal, Andhra Pradesh | Late Cretaceous | Maastrichtian | [182] |
| *Pycnodus* | Pycnodontidae | India | Lameta Formation (Madhya Pradesh) | Late Cretaceous | Maastrichtian | [177,178] |
| *Pycnodus bicresta* | Pycnodontidae | India | Asifabad | Late Cretaceous | Maastrichtian | [168,175,184,185] |
| *Pycnodus lametae* | Pycnodontidae | India | Nagpur | Late Cretaceous | Maastrichtian | [183] |
| *Pycnodus lametae* | Pycnodontidae | India | Asifabad | Late Cretaceous | Maastrichtian | [183,184] |
| *Pycnodus lametae* | Pycnodontidae | India | Rangapur | Late Cretaceous | Maastrichtian | [183] |
| *Pycnodus* cf. *P. praecursor* | Pycnodontidae | India | Asifabad | Late Cretaceous | Maastrichtian | [183] |
| *Coelodus toncoensis* ** | Pycnodontidae | Argentina | Yacoraite Formation | Late Cretaceous | Maastrichtian | [223–227] |

| Taxa | Family | Country | Locality | Period | Stage | Ref. |
|---|---|---|---|---|---|---|
| Pycnodontiformes indet. ** | Family incertae | Argentina | Yacoraite Formation | Late Cretaceous | Maastrichtian | [230] |
| *Coelodus toncoensis* ** | Pycnodontidae | Bolivia | El Molino Formation, Paja Patcha | Late Cretaceous | Maastrichtian | [233–235] |
| *Coelodus* ** | Pycnodontidae | Bolivia | El Molino Formation, Paja Patcha | Late Cretaceous | Maastrichtian | [234] |
| Pycnodontidae gen. et spp. indet ** | Pycnodontidae | Bolivia | El Molino Formation, Paja Patcha | Late Cretaceous | Maastrichtian | [233,234] |
| Pycnodontidae gen. et spp. indet ** | Pycnodontidae | Bolivia | El Molino Formation, Vila Vascarra | Late Cretaceous | Maastrichtian | [236] |
| *Coelodus* | Pycnodontidae | Madagascar | Maevarano Formation, Anembalemba Member | Late Cretaceous | Maastrichtian | [241] |
| *Pycnodus jonesae* ** | Pycnodontidae | Mali | Ménaka Formation, Iullemmeden Basin | Late Cretaceous | Maastrichtian | [238,239] |
| *Pycnodus* ** | Pycnodontidae | Mali | Ménaka Formation, Iullemmeden Basin | Late Cretaceous | Maastrichtian | [238,239] |
| *Pycnodus* | Pycnodontidae | India | Jhilmili, Chhindwara District | Late Cretaceous–Paleocene | Maastrichtian–Danian | [190] |
| *Pycnodus* ** | Pycnodontidae | India | Fatehgarh Formation, Rajasthan | Late Cretaceous–Paleocene | Maastrichtian–Selandian | [188,189] |
| *Coelodus toncoensis* ** | Pycnodontidae | Argentina | Yacoraite Formation | Paleocene | Danian | [224] |
| Pycnodontidae gen. et spp. indet ** | Pycnodontidae | India | Rajahmundry | Paleocene | Danian | [168,246,247] |
| *Pycnodus* cf. *P. praecursor* ** | Pycnodontidae | India | Rajahmundry | Paleocene | Danian | [183] |
| *Pycnodus* ** | Pycnodontidae | India | Rajahmundry | Paleocene | Danian | [183] |
| *Pycnodus jonesae* ** | Pycnodontidae | Mali | Teberemt Formation, Gao Trench Basin | Paleocene | Selandian–Thanetian | [238,239] |
| *Pycnodus jonesae* ** | Pycnodontidae | Mali | Teberemt Formation, Iullemmeden Basin | Paleocene | Thanetian | [238,239] |
| *Pycnodus* ** | Pycnodontidae | Mali | Teberemt Formation, Iullemmeden Basin | Paleocene | Thanetian | [239] |
| *Pycnodus maliensis* ** | Pycnodontidae | Mali | Tamaguélelt Formation, Taoudenit Basin | Eocene | Ypresian | [238,239] |
| *Pycnodus zeaformis* ** | Pycnodontidae | Mali | Tamaguélelt Formation, Taoudenit Basin | Eocene | Ypresian | [238,239] |
| *Pycnodus* ** | Pycnodontidae | Mali | Tamaguélelt Formation, Taoudenit Basin | Eocene | Ypresian | [239] |
| *Pycnodus* ** | Pycnodontidae | USA | Tallahata Formation | Eocene | Ypresian–Lutetian | [249] |
| *Pycnodus* ** | Pycnodontidae | USA | Lisbon Formation | Eocene | Lutetian | [249] |

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
