# Peer review of "The Fossil Record and Diversity of Pycnodontiform Fishes in Non-Marine Environments"

_diversity, doi:10.3390/d16040225_

Round 1

Reviewer 1 Report

Comments and Suggestions for Authors

This is an interesting and useful review of the freshwater fossil record of pycnodontiform fishes.

I have few comments to make, apart from a few minor corrections (including a spelling mistake in the title ;-) and comments that I wrote directly in the pdf file.

One might have expected more discussion about the number of lineages that have shifted from marine to freshwater. Because the fossil record is based primarily on isolated jaw fragments, I agree it is difficult to identify whether just one or a few lineages moved to freshwater (this is what they suggest). But because the authors know these fish well, they might have an opinion whether most of the freshwater occurrences belong to distinct lineages. Do they have insights? For example, do Paleogene freshwater occurrences belong to Late Cretaceous freshwater lineages. Even just hypotheses to be tested in these would be welcome.

Author Response

Comments and Suggestions for Authors: “including a spelling mistake in the title”

Response: This has now been rectified.

Comments and Suggestions for Authors: “But because the authors know these fish well, they might have an opinion whether most of the freshwater occurrences belong to distinct lineages. Do they have insights? For example, do Paleogene freshwater occurrences belong to Late Cretaceous freshwater lineages.”

Response: We have now incorporated a new figure of a time calibrated phylogenetic tree and here it is obvious that transition to non-marine environments has occurred numerous times within pycnodont evolution and is not restricted to only a few specialized lineages. Only two species of Turbomesodon are closely related to each other that indicate that there might be a shift within this genus to freshwater environments. With the current evidence, we can say that Pycnodontidae contains the most freshwater/brackish representatives with some basal forms and mesturids making the transition as well. However, the fragmentary fossil record certainly obscures the true diversity of the non-marine pycnodont fauna and  makes reconstructing what exactly constituted this fauna a significant challenge.

Lines 44-45: Various evidence indicates that catfish should have been present since the Early or "mid" Cretaceous, but the fossil record is very weak in the Late Cretaceous. When the record began to occur, in the Late Cretaceous, these were already heavily ossified dermal bones that one might have expected to find in older sediments. As for many extant catfishes, it is possible that most of the oldest siluriformes were inhabitants of lotic environments. (I realize that the next sentence addresses in part my comment) "

Response: I do not disagree with the points being made here but I only brought up catfishes as an example of the issues with preservation in the freshwater fish fossil record. As a result, I do not think discussion on topics such as the problematic origin of catfishes is relevant here. Furthermore, the reviewer themselves acknowledge that I explained the reasons why catfishes would have a richer fossil record than other fish lineages.

Line 84: or 'freshwater' as above?

Response: I have changed it to freshwater in the revised manuscript.

Lines 103 and 107: italic

Response: The taxa names are now italicised.

Line 264: spelling

Response: I have corrected my spelling to Pycnodontiformes.

Line 528: 'Group' Check everywhere. Also for 'Formation'.

Response: I have now maintained consistent spellings of these terms throughout the manuscript.

Line 721: italic

Response: The taxa names are now italicised.

Line 772: italic

Response: The taxa names are now italicised.

Lines 832-833: italic

Response: The taxa names are now italicised.

Lines 1033-1034: abbreviation and formatting

Response: The requested changes have been made.

Line 1177: giant space within the word ‘from’

Response: The requested changes have been made.

Reviewer 2 Report

Comments and Suggestions for Authors

Dear authors.

I thank you for having concluded a work like this, which has been necessary for a long time to remove the doubt of whether the pycnodonts invaded fresh water or not. I believe that the work is extensive and integrates a large number of fossil evidences that undoubtedly leave a positive answer to this question. I have just one observation about your Figure 4; This is difficult to read, I understand that integrating all that information into a pie chart would lead to having illegible or unrepresentable slices. That's why I suggest you change this diagram to a single bar graph, where observing the apparent increasing incidence of pycnodonts in freshwater is easier to read visually.

Author Response

Comments and Suggestions for Authors: “I thank you for having concluded a work like this, which has been necessary for a long time to remove the doubt of whether the pycnodonts invaded fresh water or not. I believe that the work is extensive and integrates a large number of fossil evidences that undoubtedly leave a positive answer to this question. I have just one observation about your Figure 4; This is difficult to read, I understand that integrating all that information into a pie chart would lead to having illegible or unrepresentable slices. That's why I suggest you change this diagram to a single bar graph, where observing the apparent increasing incidence of pycnodonts in freshwater is easier to read visually.”

Response: We have made the requested change and now have made it into a single bar chart which we hope visualizes the strikingly high number of pycnodont occurrences in both the beginning and end of the Cretaceous. We would also like to say thank you to the reviewer for their kind words on our manuscript and we feel too that it is necessary to finally have a source that summarizes our current knowledge of this phenomenon within the Pycnodontiformes. We hope this will be an invaluable resource for future workers on this fascinating group of fishes.

Reviewer 3 Report

Comments and Suggestions for Authors

The authors made a review of non-marine occurrences of Pycnodontiformes throughout the Mesozoic and early Palaeogene. They noticed a spike of non-marine occurrences of pycnodont fishes in the Cretaceous, and argued that pycnodonts might have used the Late Cretaceous freshwater environments as a refuge and began to occupy marine waters after the K/Pg extinction event. The paper improves our knowledges on the evolutionary history of pycnodonts, and I highly recommend its publication in this journal. A few annotations are given directly on the attached manuscript. 

Author Response

Comments and Suggestions for Authors: “The authors made a review of non-marine occurrences of Pycnodontiformes throughout the Mesozoic and early Palaeogene. They noticed a spike of non-marine occurrences of pycnodont fishes in the Cretaceous, and argued that pycnodonts might have used the Late Cretaceous freshwater environments as a refuge and began to occupy marine waters after the K/Pg extinction event. The paper improves our knowledges on the evolutionary history of pycnodonts, and I highly recommend its publication in this journal.”

Response:  Many thanks to the reviewer for their positive review of our paper and we hope this will be a useful repository of information for future workers wanting to investigate the evolutionary history of environmental transition in pycnodonts. We felt such a literature review was long overdue on this topic and are glad that the reviewers are in agreement with us.

Line 119: Lack of full stop and 10 mm instead of 1  cm.

Response: The requested changes have been made.

Line 237: incorrect grammer.

Response: the highlighted text is now changed to “that initially were assigned to Pycnodus mantelli by Struckmann”.

Line 832: change both species name into italic type.

Response: The requested changes have been made.